# Multiparticulate Systems of Meloxicam for Colonic Administration in Cancer or Autoimmune Diseases

**DOI:** 10.3390/pharmaceutics14071504

**Published:** 2022-07-20

**Authors:** Eva Navarro-Ruíz, Covadonga Álvarez-Álvarez, M Ángeles Peña, Carlos Torrado-Salmerón, Zaid Dahma, Paloma Marina de la Torre-Iglesias

**Affiliations:** 1Department of Pharmaceutics and Food Technology, Faculty of Pharmacy, Complutense University, Plaza Ramón y Cajal s/n, 28040 Madrid, Spain; eva.navarro.ruiz@ucm.es (E.N.-R.); ctorrado@ucm.es (C.T.-S.); zdahma@ucm.es (Z.D.); 2Instituto Universitario de Farmacia Industrial, Complutense University, Plaza Ramón y Cajal s/n, 28040 Madrid, Spain; 3Department of Biomedical Science, Faculty of Pharmacy, University of Alcalá de Henares, Ctra Madrid-Barcelona Km 33600, 28805 Madrid, Spain; angeles.pena@uah.es

**Keywords:** multiparticulate system, autoimmune disease, cancer, colonic administration, meloxicam

## Abstract

The aim of this research is the development of new colonic release systems of meloxicam (MLX) a non-steroidal anti-inflammatory drug (NSAIDs) with pH and time-dependent vehicles for cancer or autoimmune diseases. The colon has a higher pH than the rest of the gastrointestinal tract (GIT) and this can be used as a modified release strategy. Eudragit^®^ polymers are the most widely used synthetic products in the design of colonic release formulations because they might offer mucoadhesiveness and pH-dependent release. Colonic delivery systems produced with pH-dependent and permeable polymers (FS-30D) or with pH-independent and low permeability polymers (NM-30D), must dissolve at a pH range of 6.0–7.0 to delay the release of the drug and prevent degradation in the GIT, before reaching the colon. The conditions prepared to simulate a gastrointestinal transit showed the CNM multiparticulate system, composed of Eudragit^®^ NM and cellulose, as the best release option for MLX with a more sustained release with respect to the other formulations. CNM formulation followed Higuchi and First-order release kinetics, thus MLX release was controlled by a combination of diffusion and polymers swelling/eroding processes.

## 1. Introduction

Colonic delivery systems aim to release the active substance in the last portion of the small intestine, without releasing it in the upper gastrointestinal tract (GIT); therefore, it is free in the first portion of the large intestine, the ascending colon, which is a window of absorption for many high molecular weight substances. Successful colonic release requires the drug to reach the ascending colon in a precise time. In addition, the loss of active ingredients should be minimized because of the enzymatic activity of the ileum or the sequestration of the drug by the stool already compacted in the distal colon. Therefore, the objectives of an oral dosage form prepared for colonic absorption are to protect the drug throughout its gastrointestinal transit to the colon, standardize the residence time at the colonic level, guarantee system recognition by the colonic mucosa and ensure a specific release zone. Hence, for a colonic release, it is important to consider the physiological properties of the colon. In general, the GIT undergoes changes in motility, content, enzyme activity and pH from the stomach to the intestine [1]. To optimize colonic release systems, several possibilities have been studied, and we have highlighted the pH and time-dependent release systems [2].

The selective release systems at the colonic level must delay the release, which constitute an interesting alternative for the administration of some drugs in a less hostile environment than the stomach or small intestine with a high enzymatic activity [3,4,5]. The long residence times in this area of the intestine are also important, as they increase the systemic absorption of some drugs, achieving a local effect [6,7,8,9].

The colon has a higher pH (6.8–7.5) than the rest of the GIT (1.2–6.8) [10,11,12], and this can be used as a modified release strategy. In previous works [13,14] some pH-dependent formulations were prepared with polymers, such as cellulose acetate phthalate (CAP), hydroxypropyl methyl cellulose phthalates (HPMCP) 50 and 55, and copolymers of methacrylic acid and methyl-methacrylate (for example, Eudragit^®^ S100, Eudragit^®^ L, Eudragit^®^ FS and Eudragit^®^ P4135F). Eudragit^®^ polymers are the most widely used synthetic products in the design of colonic release formulations because they offer mucoadhesiveness and pH-dependent release [15,16,17,18]. The ideal polymer should be able to withstand the low pH of the stomach (pH 1–3) and proximal small intestine (pH 4.5–6.0) but dissolve at the pH of the terminal ileum and colon (pH 6.5–7.5) (see Figure 1). As a result, colonic delivery systems produced with pH-dependent polymers must dissolve at a pH of 6.0–7.0 to delay the release of the drug and prevent degradation in the GIT before reaching the colon. Generally, the dissolution of polymers, such as Eudragit^®^ FS and Eudragit^®^ NM, takes place at the terminal ileum or the ileocaecal junction when the pH exceeds 7.0 and in a short span of time drug enters the large intestine [19]. Thus, formulations could have completed transit through the small bowel at 6 h [20] when the pH value achieves more than 7.0. Eudragit^®^ NM, Eudragit^®^ FS and Eudragit^®^ FS/NM combinations have been used for colonic drug delivery. The high flexibility of Eudragit^®^ NM favors compactness and better maintains the multiparticle structure in colonic delivery systems [21].

Some studies [10] of delayed and uniform release have used the Eudracol^®^, system which is based on coating the pellet with Eudragit^®^ RL/RS and Eudragit^®^ FS30D, giving a specific colonic release in a pH- and time-dependent way. To improve specificity for the colon, Naeem et al. [11] manufactured pH- and time-dependent budesonide nanoparticles for the treatment of ulcerative colitis. These nanoparticles were prepared with Eudragit^®^ FS30D and Eudragit^®^ RS100. Eudragit^®^ FS30D is a pH-dependent polymer that dissolves in environments above 7.0, while Eudragit^®^ RS100 is a time-dependent controlled release polymer with low permeability. By combining these two polymers, premature release in the upper gastrointestinal tract is effectively minimized, achieving the desired colonic release.

The pH of the colon, which varies from 6.5 to 7.5, in normal conditions, is very sensitive to alterations by diet, diseases, water intake, and microbial metabolism [12]. For example, patients with ulcerative colitis have a more acidic colonic pH compared to healthy individuals. This would result in the incomplete release of enteric-coated systems [22]. The peristaltism of the colonic segment is slow, and the content remains in the colon for a long time, therefore, the development of time-dependent colonic formulations is also interesting [23]. Time-dependent colonic release systems guide the release of the drug by the arrival times of the food after oral administration. In general, the gastric emptying time is 15–180 min, while retention in the small intestine is 3–4 h. Thus, the release of time-dependent systems is usually around 5–6 h [24]. Currently, the drug is coated with insoluble coatings that are difficult to decompose and the release time is controlled by the proportion and dose of the coating material [25]. Eudragit^®^ NM 30 D, neutral, with low permeability is another attractive option since it can be formulated with all active ingredients regardless of the ionic charge they have and in case highly flexible films are required.

MLX is BCS class II drug (meloxicam, MLX) (4-hydroxy-2-methyl-N-(5-methyl-2-thiazolyl)-2H-1,2-benzothiazine-3-carboxamide) [26]. Moreover, it is a non-steroidal anti-inflammatory drug (NSAIDs) that is distinguished by inhibiting COX-2 to a greater degree than COX-1 in both its cyclooxygenase and peroxidase activity. Because MLX is well absorbed in the colonic region and because of its properties against colon cancer and autoimmune anti-inflammatory diseases, it is interesting to study new colonic formulations of MLX [27,28]. Nevertheless, few studies have considered the development of colonic formulations using multiparticulate systems such as pellets or granules of MLX. In fact, MLX tablets containing Eudragit^®^ FS 30D, as an outer coating layer, were prepared for colon targeting [25]. The decline in pH from the end of the small intestine to the colon can also result in problems, lengthy lag times at the ileo-cecal junction or rapid transit through the ascending colon which can also result in poor site-specificity for enteric-coated single-unit formulations [3]. In addition, multiparticle systems hold great potential for enhancing drug targeting as well as drug uptake. Therefore, as shown by several authors, various formulations with particle size reduction may be beneficial for colon-targeted drug delivery [29], such as MLX pellets with Eudragit^®^ FS [30] and MLX polymeric micelles based on amphiphilic chitosan derivatives [31]. The pellets production process is very difficult to control, and also very costly, meanwhile polymeric micelles present temporal stability and additional concerns might be linked to their drug loading capacity. However, the development of MLX granulates could be a good candidate for colon targeting because of their simple manufacturing process, good stability, homogeneous drug load and could provide site-specific drug delivery [32].

The prevalence of colonic diseases has increased worldwide, demanding the effectiveness of local treatments for more effective and safer therapies. Of all colonic diseases, colorectal cancer (CRC) causes the highest number of cancer deaths in Europe (with more than 200,000 deaths annually and is the third most diagnosed cancer in the world [33]. The incidence of inflammatory bowel disease (IBD) is also growing alarmingly in areas such as Asia, where there was previously low incidence. Consequently, the need for effective treatment for colonic diseases is one of the world’s public health problems.

This work explores MLX-colonic release systems that would be a good proposal for the treatment of colonic diseases, thanks to its anti-inflammatory effect. Its usefulness in chemoprevention, chemo-suppression and UV protection has recently been studied [27,28]. The purpose of this research is the development of new colonic release systems of MLX with pH and time-dependent vehicles, avoiding pH degradation across the GIT, for cancer or autoimmune diseases.

## 2. Materials and Methods

### 2.1. Materials

Meloxicam (MLX, Fagron Ibérica, Barcelona, Spain); Methylcellulose (Metolose^®^ 90 SH 100, Shin-Etsu Chemical Co., Ltd., Tokyo, Japan); aqueous dispersion of a copolymer of ethyl acrylate and methyl methacrylate (Eudragit^®^ FS 30D and Eudragit^®^ NM 30D, Evonik Röhm GmbH, Wörth am Main, Germany); α-lactose monohydrate (Fagron Ibérica, Barcelona, Spain). All other chemicals were of analytical grade.

### 2.2. Preparation of Formulations

We previously described the development of these colonic granulates [32], and in the present study, we reformulate them but select different particle sizes. Briefly, the studied MLX was granulated with lactose and two types of Eudragit^®^: FS and NM; some granules containing 5% of cellulose (Metolose^®^ 90 SH 100). The granules are made in four successive granulations, leaving a day of drying between each one, until approximately 30% of Eudragit^®^ is added to the different formulations. A Blank formulation, with the four successive granulations, without MLX, and containing lactose, Metolose^®^ and Eudragit^®^ NM was manufactured for their studying in either scanning electron microscopy (SEM), X-ray powder diffractometry (XRPD) or grazing incidence X-ray diffraction (GID), respectively. This granulation process allows for obtaining less porous formulations, compared to other processes such as lyophilization [34]. A summary of this process is shown in Figure 2.

Finally, different diameters were selected by sieving to determine their in vitro behavior at different pH buffers, in dissolution studies (apparatus I), which are presented in Table 1. The sieving method is a conventional and quantitative method for particle size determination. It, however, requires a high amount of the sample under investigation for each measurement [35].

After the dissolution rate data obtained for MLX, the most suitable formulation would be selected and prepared by wet granulation of the MLX with 30% Eudragit^®^ FS or NM alone, or with a combination of both polymers NM + FS (1:1), hereinafter NM + FS, either adding a 5% cellulose (Metolose^®^) obtaining CFS, CNM or CNM + CFS (1:1), hereinafter CNM + CFS. Subsequently, the physicochemical characterization of the best formulation in the in vitro tests was carried out using SEM, XRPD or GID analytical techniques.

### 2.3. Solubility Study

Solubility tests of the granules (FS, NM, NM + FS, CFS, CNM, CNM + CFS) were carried out with 3 mg of MLX or its equivalent amount in 0.5–1.41 mm diameter size granules, in 2 mL of pH 6.8 buffer in a thermostatic bath at 37 ± 1 °C stirring for 2 days. Samples were filtered and diluted 1/50 with a 6.8 buffer to be quantified. The cumulative amount of MLX released from the formulations was determined at 362 nm with a UV-VIS JASCO V-730 spectrophotometer (Jasco^®^ International Co., Ltd.; Tokyo, Japan), with the following calibration curve: y = 0.0508x (µg/mL) + 0.0105 (r^2^ = 0.9995) across a range of 2–15 µg/mL. Each determination at each time was performed in triplicate, and the error bars on the graphs represent the standard deviation.

### 2.4. In Vitro Release Profile Study

To evaluate MLX in the different granulations, a quick and simple analytical method has been developed: a UV spectrophotometry technique. Different wavelengths were selected to carry out the normal MLX spectrophotometric test according to the pH study, taking the wavelength at which, the MLX presented a maximum absorption in the different pH medium. At pH 7.4, the cumulative amount of MLX released from the formulations was determined at 361 nm with a UV-VIS JASCO V-730 spectrophotometer (Jasco^®^ International Co., Ltd.; Tokyo, Japan), with the following calibration curve: y = 0.0512x (µg/mL) + 0.0141 (r^2^ = 0.9995) across a range of 2–15 µg/mL. At pH 6.8, the MLX released from the formulations was determined at 362 nm by the spectrophotometric method described in the solubility study. Then at pH 1.2, the cumulative amount of MLX released from the formulations was determined at 346 nm with a UV-VIS JASCO V-730 spectrophotometer (Jasco^®^ International Co., Ltd.; Tokyo, Japan), with the following calibration curve: y = 0.0377x (µg/mL) + 0.0368 (r^2^ = 0.9999) across a range of 1–15 µg/mL. The determination at each time point was performed in triplicate, and the error bars on the graphs represented the standard deviation. Dissolution studies were performed using the United States Pharmacopeia (USP) basket method (apparatus 1) in ERWEKA^®^ DT 80 (ERWEKA GmbH; Langen, Germany) dissolution equipment. The temperature was maintained at 37.0 ± 0.1 °C throughout the dissolution study.

Dissolution rate at pH 6.8: The first dissolution rate test (apparatus I) was carried out with the 0.50–1.41 mm granules, which contained 10 mg of MLX. The vessels contained 900 mL of pH 6.8 phosphate buffer, at 37 ± 0.1 °C, with a rotation speed of 50 rpm. Taking samples at 5, 15, 30, 45, 60, 120, 180, 240, 300, 360 and 390 min. The samples were filtered through 0.45 µm (Millipore^®^) and analyzed directly.

The dissolution rate of the granules of 0.350–1.50 mm was studied at pH 6.8, 37.0 °C ± 0.1 and 50 rpm. The test lasts 8 h, and the samples were taken at 5, 15, 30, 45, 60 min and then every hour until 8 h. Samples were filtered through 0.45 µm and analyzed at 362 nm.

The dissolution test in pH 7.4-phosphate buffer, with granules of 0.350–1.50 mm, at 50 rpm and 37.0 °C ± 0.1. Reading the samples at 5, 15, 30, 45 and 60 min and then every hour until 8 h, then filtering and analyzing at 361 nm wavelength.

The last dissolution test was carried out at pH 1.2, 6.8 and 7.4 with buffer changes: A test was carried out starting from 900 mL of pH 1.2 in the apparatus I, at 37 ± 0.1 °C and with the granules of size 0.85–1.00 mm, at a speed of 50 rpm. It was made with CFS, CNM and CNM + CFS granules. Samples were taken at 1 h and two hours later, we removed and changed the medium trying not to lose any granules, filling up to 900 mL with pH 6.8 buffer. Samples were then taken every half hour, at 2.5, 3, 3.5, 4, 4.5 and 5 h, to change the medium at 5 h, we changed the medium to pH 7.4, taking samples at 5.5, 6, 6.5, 7, 7.5 h, and finally at 24 h. Samples were filtered at 0.45 µm.

To investigate the effect either of different particulate diameters or diverse polymers on the release of MLX more precisely, the results were analyzed according to the Korsmeyer–Peppas Equation for *M_t_*/*M*_∞_ < 0.6, which can be expressed as the following Equation:*M_t_*/*M*_∞_= *K_d_*·*t^n^*(1)
where *M_t_*/*M*_∞_ is the fractional drug released at time *t* (h) from the total amount released *M_∞_*·*K_d_* (min^−^^1^) is the kinetic dissolution constant, and *n* is a diffusional exponent characteristic of the release as a function of time *t*. For drug release from tablets, which contain hydrophilic polymers, the exponent *n* is the diffusional constant that characterizes the drug release transport. When *n* = 0.5, a Fickian diffusion process was observed, where drug diffusion through the polymeric matrix is the dominant release mechanism.

When *n* values are between 0.5 < *n* < 1, drug diffusion occurs via anomalous transport (non-Fickian). In anomalous diffusion, it is assumed that the mechanism of CL release is a combination of swelling, erosion, and diffusion. When *n* = 1, Case II transport or zero order release kinetics could be observed. An *n* value of about 1.0 indicates that polymer relaxation, polymer dissolution, or tablet erosion are the dominant mechanisms [36]. For the mathematical evaluations, we characterized drug release kinetics by fitting standard release to Higuchi, zero-order and first-order models [37].
*M_t_*/*M_∞_* = *K*_0_·*t* zero-order model(2)
*M_t_*/*M_∞_* = *K_H_*·*t*^0.5^ Higuchi model (3)
*Ln* (*M_t_*/*M_∞_*) = −*K*_1_·*t* first-order model (4)
where *M_t_*/*M_∞_* is the fractional drug released at time *t* (min) from the total amount released *M_t_*·*K_H_*, *K*_0_, and *K*_1_ (min^−1^) are the kinetic dissolution constants for zero-order, Higuchi, and first-order kinetic models, respectively, which characterize release as a function of time *t*.

The high r^2^ values in the zero-order model indicate that polymer relaxation or tablet erosion is the dominant mechanism and is related to Case II transport for the Korsmeyer–Peppas Equation. The high r^2^ values in the Higuchi model indicate the better fit of release data for diffusion kinetics, which are related to a Fickian diffusion process in the Korsmeyer–Peppas equation. The high r^2^ values in the first-order model indicated swelling and erosion phenomena and could be related to non-Fickian (anomalous transport) for the Korsmeyer–Peppas equation.

### 2.5. Scanning Electron Microscopy (SEM)

The samples were mounted on an aluminum sample mount. After coating with a thin layer of gold-palladium, the hydrogel samples were analyzed via SEM using a Jeol^®^ 6400. All micrographs were the product of secondary electron imaging used for surface morphology identification at different magnifications and an accelerating voltage of 20 kV.

Samples of CNM, CFS and CNM + CFS multiparticulate systems during the dissolution test after 2 h at pH 1.2, 3 h at pH 6.8 and 3 h at pH 7.4 (2, 5 and 8 h) were withdrawn from the dissolution medium and lyophilized, then they were analyzed by SEM.

### 2.6. X-ray Powder Diffractometry (XRPD) and Grazing Incidence X-ray Diffraction (GID)

XRPD was used for characterizing crystalline materials and providing information on structures. The dust and surface sweep method were used. Approximately 20 mg of sample was dispersed on a sample holder and loaded into a Philips X’PERT^®^ diffractometer equipment. Each sample was exposed to Cu-kα radiation, the angles of incidence being from 5° to 50°. The test conditions are 40 mV voltage and 55 mA intensity. The diffractograms obtained from the raw materials (MLX, lactose, Metolose^®^ and Eudragit^®^) and the CNM formulation and its placebo make it possible to know if the samples are crystalline or amorphous. GID method has also been used, which consists in that the angle of incidence (θ) remains fixed while the detector moves normally around the axis of the goniometer [38]. GID is a surface-sensitive technique that, allows the characterization of epitaxial films with thicknesses down to a few atomic layers. All measurements were made with a Philips X’PERT^®^ diffractometer, where the radiation was the same as for the measurements by the powder method. In this study, we have used an incidence angle (θ) of 1°. Before the measurements, the equipment was carefully aligned and calibrated. A spatial sample holder was used that could be adjusted to the height of the sample. Under these conditions, the CNM formulation was measured.

### 2.7. Statistical Analysis

Differences between obtained values (Mean ± SE) for the prepared formulae and the control formulae were carried out using a one-way analysis of variance (ANOVA), followed by an appropriate post-hoc test in the case of the presence of a significant difference. A *p*-value less than 0.05 was considered a criterion for a statistically significant difference.

## 3. Results and Discussion

### 3.1. Solubility Studies at pH 6.8 with MLX Raw Material and 0.50–1.41 mm Granules of Formulations: FS, NM, NM + FS, CFS, CNM and CNM + CFS

The purpose of this study is to evaluate the influence of Metolose^®^ and different Eudragit^®^ polymers NM or FS on MLX solubility properties, which could affect its dissolution performance, thus affecting its colonic bioavailability [39].

As can be seen in Figure 3, the solubility of MLX raw material at this pH is 249.02 ± 10.44 µg/mL. The drug solubility coefficient is only slightly improved in the formulation with CFS granules (1.09-fold) compared to pure MLX. In all the other multiparticulate systems, it is decreased compared to the pure MLX. The FS formulation is the one with the lowest solubility (99.90 ± 1.39 µg/mL). CFS and CNM multiparticulate systems with Metolose^®^ showed a significant increase (*p* < 0.05) in solubility, with 2.71 and 1.38-fold increases for CFS and CNM, respectively, compared to FS and NM. Metolose^®^ seems to help granules disintegrate, achieving higher dissolution values [40]. Meanwhile, in CNM + CFS formulation Metolose^®^ did not have any influence over MLX solubility. Probably, Eudragit^®^ FS together with Eudragit^®^ NM form a more cohesive film coat, avoiding water absorbance of Metolose^®^ [41].

In other studies, Weyna et al. [42] analyzed the solubility of MLX at pH 6.5 with respect to co-crystals of the active ingredient, correlating the results in vitro with in vivo tests performed in rats, administering a suspension with the equivalent amount of 10 mg/kg of MLX. It was shown that samples with a greater solubility at that pH also had higher absorption and an earlier onset of action, so it is expected that the granules with a solubility coefficient higher than MLX raw material will have a higher absorption in vivo. In addition, Patel et al. [25] reported that this increase in the solubility of the MLX at this pH (small intestinal pH), which leads to an increase in absorption from the small intestinal region.

In the solubility studies, the formulations with Metolose^®^ experienced an increase in solubility, with the exception of the CNM + CFS granulate. It was decided to verify these results by performing a dissolution rate test in a pH 6.8 medium, at 37 °C for pure MLX and all formulations: FS, NM, NM + FS, CFS, CNM and CNM + CFS.

### 3.2. In Vitro Release Profile Study

This study is important to evaluate the influence of Metolose^®^ and different Eudragit^®^ polymers NM or FS on MLX dissolution performing at different pHs simulating the gastrointestinal environment, which could affect its colonic bioavailability [39]. To investigate the effect either of different particulate sizes or of diverse polymers on the release of MLX more precisely, the results were analyzed according to the Korsmeyer–Peppas Equation. In addition, it characterized drug release kinetics by fitting standard release to Higuchi, zero-order and first-order models.

#### 3.2.1. First Dissolution Tests at pH 6.8 Were Carried out with 0.5–1.41 mm Granules for FS, NM, FS + NM, CFS, CNM and CNM + CFS Samples

The use of time-dependent polymers (Eudragit^®^ NM) and pH-dependent polymers (Eudragit^®^ FS) and the use of combinations of Eudragit^®^ time- and pH-dependent polymers have been used in different works to improve colonic release [43,44]. Moreover, the absence of burst effects in the different Metocel^®^/Eudragit^®^ granule justifies the efficient interpenetration of MLX within the network [45].

As can be seen in Figure 4, CNM granules achieved the maximum percentage released of MLX at 8 h (88.77 ± 3.46%), yielding more than the reference active ingredient, the pure MLX (78.10 ± 3.01% at the same time), followed by the CFS, with 66.34 ± 2.12% and CNM + CFS with 61.86 ± 2.60% at 8 h. Different studies have shown how the improvement in drug solubility in a pH zone (6.5–7) increases drug solubility at the colon site [25,46]. CNM formulation had the lowest “burst effect”, thus retaining better MLX for possible enteral use. As expected, the lowest results were obtained for samples without the disintegrating material (Metolose^®^), FS, NM and FS + NM.

Based on the results obtained, the CNM granules were selected for further studies, followed by the CFS, and CNM + CFS, since it is at this pH (colonic) that we would be interested in starting the release of the MLX. Han and Choi [47] studied the pharmacokinetics of MLX with ethanolamine salts through their dissolution profiles at pH 1.2 and 6.8 in vitro and in vivo tests in rats, showing that the formulations with the higher dissolution at pH 6.8 also had a faster absorption of the MLX in rats, so it was expected for the CNM granulate. Therefore, the granules, CFS, CNM and CNM + CFS were prepared for a dissolution rate study at different pHs, in an in vitro simulated gastrointestinal environment.

To evaluate the influence of MLX within the multiparticulate systems, drug release kinetics were determined for the different formulations. These granules are composed of a matrix based mainly on lactose, a hydrophilic diluent, and in some systems, it is also made up of a proportion of Metolose^®^, disintegrant, and they are coated with, either Eudragit^®^ NM, Eudragit^®^ FS or a combination thereof. The drug may interact differently with the various components of the matrix. Table 2 shows the Korsmeyer–Peppas, Higuchi and first-order kinetic models of formulations: FS, NM+FS, NM, CNM + CFS and CNM. The Korsmeyer-Peppas kinetic model exhibits anomalous (non-Fickian) dissolution with n values ranging from 0.60 to 0.82, for most formulations excepting NM + FS, NM and CFS, suggesting that the rate of water uptake into the matrix of granules and MLX release were controlled by diffusion through the system structure and dissolution and swelling/erosion processes [48]. The absence of burst effects in both, CNM and CNM + CFS granules, justified the efficient interpenetration of MLX within the network [45]. According to the Korsmeyer–Peppas model, the release rate and release mechanism for CFS granules are governed by a complex quasi-Fickian release mechanism [49]. Moreover, for both systems NM + FS and NM, diffusional exponent (n) values greater than 0.89 characterize Super Case II transport [50] which is controlled by polymer chain relaxation and erosion.

Not all formulations fit the zero-order kinetic model (data not shown). However, the first-order kinetic model of multiparticulate colonic systems with Eudragit^®^ NM showed the highest r^2^ values, and CNM formulation was the best fitting granulate (0.995), followed by the Higuchi kinetic model (best r^2^ values for Eudragit^®^ NM systems), as presented in Table 2. These results indicate that MLX release is controlled by a combination of diffusion, together with rapid swelling and erosion processes from the lactose, MC and Eudragit^®^ NM matrix. These multiparticulate colonic systems showed kinetics of MLX dissolution that fit non-Fickian transport rather than Case II transport [48].

#### 3.2.2. Dissolution Rate of Granules of Size 0.35–1.50 mm at Different pHs (6.8 and 7.4)

Time-dependent polymers with Metolose^®^ such as CNM and CNM + CFS could have less influence on the microenvironment pH induced by the solubilized MLX, which is a key factor controlling drug release [32]. However, the solubilization of Eudragit^®^ FS is pH dependent, the effect of the micro pH environment created by the drug could affect their dissolution profiles [46].

Afterward, the widest range of particle diameter was used (0.350–1.50 mm) for the selected formulations: CFS, CNM and CNM + CFS, comparing them to MLX raw material, were analyzed separately at pH 6.8 (Figure 5) and pH 7.4 (data not shown).

At pH 6.8, it was evident that formulations with Eudragit^®^ NM did not exhibit a burst effect. CNM granulate at 8 h managed to release something more (98.48 ± 2.28%) than the pure MLX (96.14 ± 10.76%) and yielded in a more sustained manner than the rest of the formulations (Figure 5), having a similar profile to the CNM + CFS granules. However, at 180 min CNM + CFS system released slightly more MLX (59.23 ± 8.04%) than CNM formulation (56.19 ± 4.24%). The granulate that released the most MLX at initial times was CFS, achieving a 44.02 ± 0.55% and 75.90 ± 1.44% at 30 min and 60 min, respectively, but also the CFS profile was uncontrolled and possessed a great burst effect. Following oral administration, these systems are subjected to challenges from the harsh and hostile environment of the GI tract before reaching the colon. For instance, the unwanted burst drug release in the stomach and the small intestine before reaching the colon is an intrinsic limitation of colon-targeted drug delivery systems [51]. This may be due to both factors: disintegrating effect of Metolose^®^ which, between the intermolecular spaces of the cellulose allows the passage of water and, with it, a greater release of drug [52] and using Eudragit^®^ FS, a pH-dependent polymer, which allows drug release to be reduced at pH greater than 7.0 [46]. These data indicated that a high release of MLX from the CFS formulation at pH 6.8 decrease its chance of success in reaching the colon and exerting its action there. We are looking for a formulation that releases in the colon, so we could select the formulation that contains a time-dependent Eudragit^®^, which delivers at the desired level of the GIT.

Table 3 shows the Korsmeyer–Peppas, Higuchi and first-order kinetic models of formulations: CFS, CNM + CFS and CNM. The Korsmeyer–Peppas kinetic model for the three formulations, with this large range of particulate size, exhibits *n* values of about 1.0, indicating that polymer relaxation and polymer dissolution are the dominant mechanisms, which agrees with Eudragit^®^ FS dissolution characteristics [53]. The Higuchi kinetic model of the CNM multiparticulate colonic system showed the highest r^2^ values, followed by the first-order kinetic model, as presented in Table 3. The kinetic models’ results revealed that MLX release is controlled by a combination of diffusion (major), together with dissolution and erosion processes from Metolose^®^ and Eudragit^®^ (minor). The wide particle size range slightly affects the fit of the MLX release to the different kinetic models, as seen in Table 2 and Table 3.

Assay carried out in pH 7.4 medium (data not shown) exhibited that the CFS granulate with pH dependent and permeability characteristics continues to be the formulation that yields faster the MLX followed by the combination CNM + CFS. On the other hand, the CNM granulate, which shows a low permeability and pH independence, achieved a sustained release profile that is adequate for an enteral administration.

The kinetic parameters of these multiparticulate colonic systems at this pH (7.4) could not be evaluated due to their fast dissolution.

#### 3.2.3. Dissolution Rate of MLX at pH 1.2, 6.8 and 7.4, of 0.85–1.00 mm Diameter Granules of CFS, CNM and CNM + CFS with Buffer Changes (Simulating the Gastrointestinal Environment)

The use of dissolution studies with pH changes at 1.2, 6.8 and 7.4 are more suitable to determine the dissolution rate of drug colon targeting systems [43,44]. Finally, the selected multiparticulate colonic systems were tested in three different media of pH 1.2, 6.8 and 7.4, for simulating their transit through the gastrointestinal tract.

At pH 1.2 (see Figure 6A), it was observed that MLX raw material was the only sample that began to dissolve, reaching a 3.63 ± 1.98% at 2 h of testing. Therefore, the other formulations do not begin to release at the pH of the stomach and may reach other parts of the gastrointestinal tract intact. The results show that at 8 h, in this simulated model of the gastrointestinal tract, CNM granulate continues to be the one that has yielded the most, with 90.32 ± 2.34%; but, after passing through the acid medium, the next best is the CNM + CFS, with 52.42 ± 3.25% and finally the CFS (32.24 ± 4.83%) contrary to what happened when the dissolution rate was determined only at pH 6.8. On the other hand, the size of the granules affected their dissolution, achieving a higher percentage of MLX dissolved at 24 h in the granules that presented a greater specific surface [54] except in the case of the CFS granule, but the order of dissolution of the multiparticulate systems was not altered at 8 h, since it was still the CNM (90.32 ± 2.34%) that released the most MLX at that time, compared to the pure MLX with a 46.96 ± 4.91% achieved at 8 h. These results showed that CNM formulation continues to present a low burst effect, possessing a sustained release, which could allow a high release of the active ingredient at the pH of the site of action and at times consistent with intestinal transit. In addition, after successive filtrations and medium changes, it was observed that the three formulations (CFS, CNM and CNM + CFS multiparticulate colonic systems) maintained their full structure, as granules.

To evaluate the influence of MLX within the granulate matrix, drug release kinetics were determined for the different formulations. Figure 6B,C show the Higuchi and first-order kinetic models, respectively, for multiparticulate systems: CNM, CNM + CFS and CFS. Because of the pH changes, these results could not be fitted to the Korsmeyer–Peppas kinetic model.

CNM formulation presented highest r^2^ values in the first-order (0.9661) and in Higuchi (0.9961) kinetic models, respectively (Figure 6B,C); followed by CNM + CFS (0.9651 and 0.9729) and finally CFS obtained the worst results for both first-order (0.9585) and Higuchi (0.9476) kinetics, respectively. In view of these results, we can conclude that in formulations containing Eudragit^®^ NM, a time-dependent polymer, and Metolose^®^, the predominant mechanism of drug release is diffusion [55] until the Eudragit^®^ NM polymer becomes more porous and dissolves. From then on, the phenomenon of swelling and erosion of the Metolose^®^ begins to occur, which controls the release of MLX to a lesser extent.

### 3.3. Scanning Electron Microscopy (SEM)

The Scanning Electron Microscopy (SEM) method was used to study the surface and morphological characteristics of different samples: Metolose^®^, Eudragit^®^ FS and Eudragit^®^ NM in their pure states, MLX raw material, CNM granulate (selected for its adequate dissolution rate characteristics) composed of MLX, lactose, Metolose^®^ and Eudragit^®^ NM, depending on the sample, and its blank formulation (which had lactose instead of the active ingredient) with four successive granulations. Moreover, we have performed SEM surface images which shall give information about changes in the surface morphology during dissolution [56], simulating the gastro-intestinal conditions: pH 1.2 (gastric pH), pH 6.8 (small intestine pH) and pH 7.4 (colon pH).

Scanning electron micrographs of MLX raw material at different magnifications (data not shown) exhibit the morphological characteristics of the drug, whose structure is in the form of recognizable cubic crystals with a smooth surface [57]. This MLX characteristic structure can be well observed in the multiparticulate system with four granulations analyzed at a magnification of 200× (Figure 7A). However, MLX was not seen, since the drug is not present, in the blank granules, as in the case of the blank formulation with four successive granulations (Figure 7B).

This detail is better appreciated in the micrographs obtained with a magnification of 1500× (Figure 7C,D), verifying that MLX is not seen in the blank sample (Figure 7D), observing the presence of lactose only, easily distinguishable for its large crystals. These changes were also observed in other types of blank formulations [58].

In the micrographs obtained at a magnification of 20× (Appendix A), it was also possible to identify the approximate particle morphology, size, and shape by SEM. Benavent C et al. [59] performed an SEM analysis identifying its surface and an approximate particle size range for different solid dispersions. In Appendix A (CNM formulation), irregular particles with a matrix structure and a rough surface (0.51–0.24 mm) can be seen, and in Appendix A (blank CNM formulation) similar structures are shown with a slightly larger particle size (0.67–0.27 mm).

In Appendix A can be seen the surface morphology of Metolose^®^, Eudragit^®^ FS and Eudragit^®^ NM in their pure states. As seen in Appendix A, methylcellulose (Melose^®^) appears as irregular flake fibers, also described by other authors [56]. On the other hand, Eudragit^®^ FS and NM displayed a very smooth film surface, with some scales of different thicknesses, this smooth surface has been previously reported in formulations with Eudragit^®^ [60].

Appendix A displays scanning electron micrographs of the surfaces of multiparticulate systems CFS, CNM + CFS and CNM, after 2 h at pH 1.2; 3 h at pH 6.8 and 3 h at pH 7.4. After 2 h in simulated gastric fluid (pH 1.2) the surface structure of all three formulations remained very similar to the initial time (t_0_), presenting a very rough surface (see Figure 7). However, after being in an alkaline medium (pH 6.8), the three multiparticulate systems showed a more porous structure. Similar studies were carried out by several authors [53,56,61]. Therefore, SEM images (pH 6.8) let us assume that the dissolution happens through pores that are generated by a combination of drug diffusion [56] and polymer swelling/eroding processes [53]. Meanwhile, after being 3 h in pH 7.4 CFS and CNM + CFS systems showed a smoother structure, this could be related to a rapid polymer disintegration. In contrast, the CNM formulation still had a rough surface, maintaining its structure. This result may be related to its more controlled dissolution rate. Similar results were obtained by Mehta et al. [62] in Eudragit^®^ S-100 coated naproxen matrix tablets for colon-targeted drug delivery system.

The SEM results allowed us to consider that the granulation process managed to effectively create a consistent polymeric matrix structure in CNM formulation, thus achieving a sustained release throughout the gastrointestinal tract.

### 3.4. X-ray Powder Diffractometry (XRPD)

The objective of X-ray powder diffractometry (XRPD) was to identify the changes in the crystalline structure of the MLX raw material and the possible dilution effect due to the addition of different excipients such as lactose, Eudragit^®^ and Metolose^®^ incorporated into the formulation CNM. Meanwhile, the grazing incidence X-ray diffraction (GID) was to perform a better evaluation of the surface of the selected formulation (CNM) due to its adequate MLX release profile [38].

Figure 8A shows the differences between the diffractograms obtained by XRPD and GID techniques of the CNM granulate containing MLX. The classical XRPD method shows the main characteristic peaks of MLX in the CNM granulate. Likewise, when using the GID technique, it is also possible to distinguish the majority peaks of the MLX at angle 2 (θ) positions of 8.5 ± 0.2° and 14.5 ± 0.2°. In contrast, as expected, the XRPD diffractogram of the CNM placebo pellet does not show any peak characteristic of MLX. In addition, it is shown that the pure MLX is presented, in its polymorphic form III, which is characterized by peaks at the positions of 8.5 ± 0.2°, 10.7 ± 0.2°, 12, 7 ± 0.2° and 14.54 ± 0.2° [63] and characteristic peaks at 16.4 ± 0.2° and 17.4 ± 0.2° for lactose [64]. However, with the GID technique in the CNM formulation, a peak appears at position 28.4 ± 0.2° with an elevated intensity, which corresponds to a particular crystalline component of the granulate with large crystals, possibly due to lactose. In all samples analyzed by the classic XRPD technique, except for the MLX raw material, an identical peak is seen in this position, regardless of whether they contain MLX. To identify that peak, it is decided to analyze all the excipients of CNM formulation (see Figure 8B), to try to identify it. Thus, lactose, Metolose^®^ and Eudragit^®^ NM were analyzed. The latter was previously allowed to dry in a petri dish and then pulverized to be assayed by the XRPD technique. Diffractograms of Metolose^®^ and Eudragit^®^ NM presented halos that can be associated with a semi-crystalline state but are mostly amorphous. Both polymers Metolose^®^ and Eudragit^®^ could present a modest degree of crystallinity that has been previously described [65,66]. Lactose presented a highly crystalline pattern, with a peak at the same position (28.4 ± 0.2°) as in both XRPD and GID diffractograms of CNM formulation. The XRPD results of the CNM formulation demonstrated that after the granulation process, the original structure of all components is maintained.

## 4. Conclusions

Different multiparticulate MLX systems have been designed for colonic administration based on lactose, methyl cellulose (Metolose^®^), Eudragit^®^ NM, FS and combinations of both polymers. CFS and CNM multiparticulate systems with Metolose^®^ showed a significant increase (*p* < 0.05) in solubility, compared to FS and NM formulations. Under conditions such as gastrointestinal transit (pHs 1.2, 6.8 and 7.4), the CNM formulation showed the best MLX release characteristics, with a more sustained profile compared to the other formulations studied. CNM multiparticulate system presented the highest r^2^ values in both the first order (0.9661) and Higuchi (0.9961) kinetic models, respectively, followed by CNM + CFS (0.9651 and 0.9729), and finally, CFS obtained the worst results for both first order (0.9585) and Higuchi (0.9476) kinetics, respectively. In view of these results, we can consider that in formulations containing Eudragit^®^ NM, a time-dependent polymer, and Metolose^®^, the predominant mechanism of drug release is diffusion followed by swelling/eroding phenomena. SEM results showed that the granulation process managed to effectively coat the MLX in CNM granulate, thus achieving a sustained release throughout the gastrointestinal tract. The X-ray diffraction results of the CNM formulation demonstrated that after the granulation process, the original structure of all components is maintained. Finally, colonic release formulation of MLX has been achieved, which reaches the large intestine mostly intact, with a sustained release along the intestinal tract due to the use of Eudragit^®^ NM in four successive granulations and a high release of active ingredient at the site of action, due to the presence of Metolose^®^. It also has an adequate shape and size, suitable for uniform dosage and possibly appropriate for the treatment of diseases involving colonic inflammation such as ulcerative colitis, Crohn’s disease, and colon cancer, among others.

## Figures and Tables

**Figure 1 pharmaceutics-14-01504-f001:**
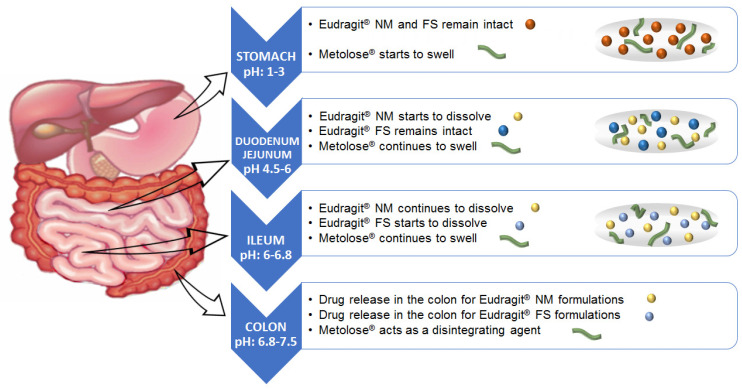
Time-dependent polymer, pH sensitive polymer and cellulose polymer behavior in the gastrointestinal tract.

**Figure 2 pharmaceutics-14-01504-f002:**
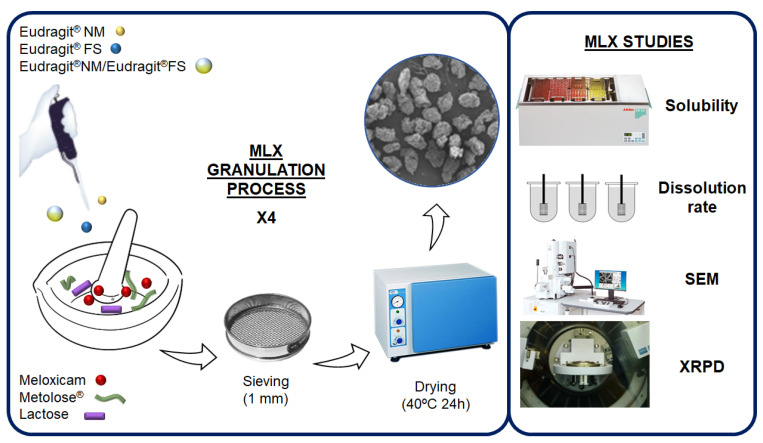
Preparation procedure for MLX multiparticulate colonic systems and their characterization.

**Figure 3 pharmaceutics-14-01504-f003:**
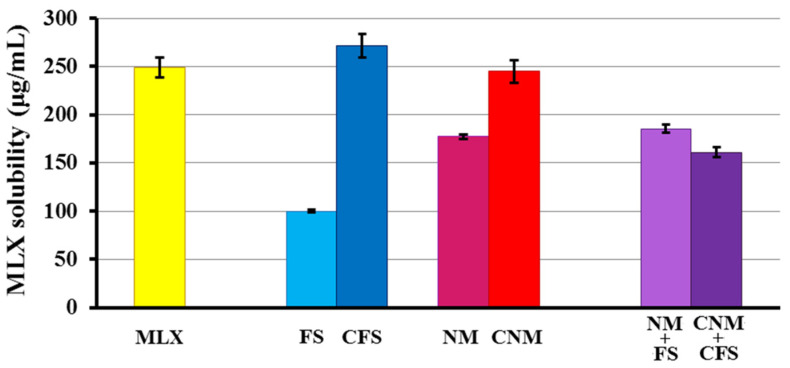
Solubility study of MLX raw material, and multiparticulate colonic systems: FS, NM, NM + FS, CFS, CNM and CNM + CFS in phosphate buffer (pH 6.8).

**Figure 4 pharmaceutics-14-01504-f004:**
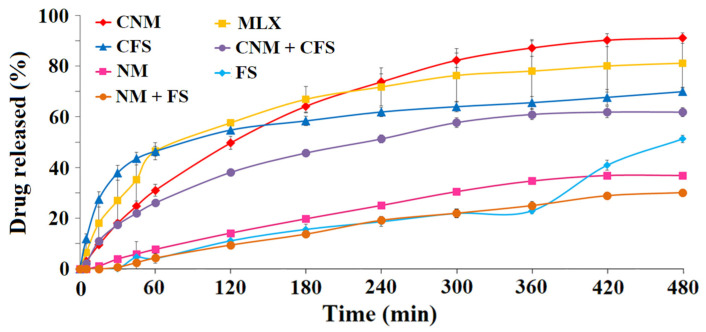
Release profiles at pH 6.8 for MLX raw material and multiparticulate colon systems of 0.50–1.41 mm diameter size: FS, NM, NM + FS, CFS, CNM and CNM + CFS.

**Figure 5 pharmaceutics-14-01504-f005:**
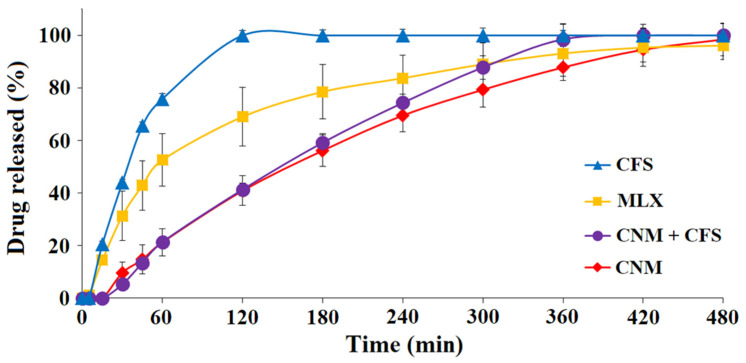
Release profiles of multiparticulate colonic systems of 0.35–1.50 mm diameter size (CNM, CFS and CNM + CFS) and pure MLX in pH 6.8 medium.

**Figure 6 pharmaceutics-14-01504-f006:**
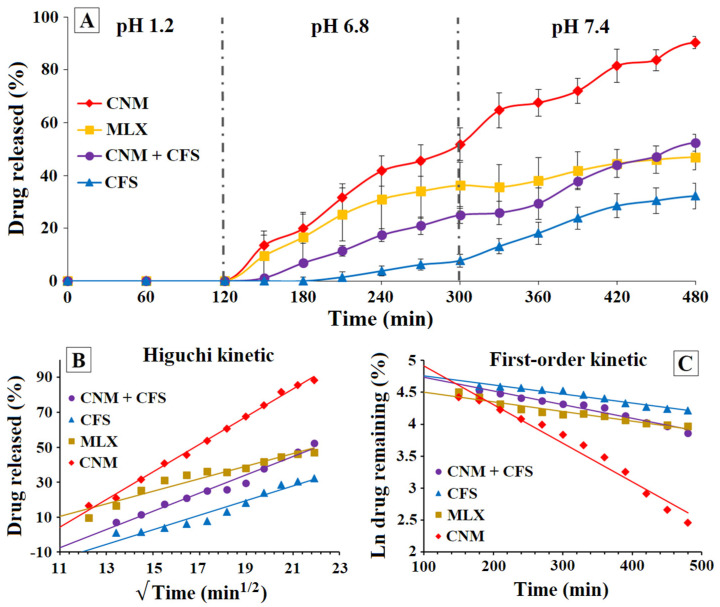
(**A**) Dissolution rate of MLX with buffer changes (pH 1.2, 6.8 and 7.4) for MLX raw material and 0.85–1.00 mm size diameter granules of CNM, CFS and CNM + CFS. (**B**) Higuchi kinetic model applied for MLX release from MLX raw material, multiparticulate colonic systems CNM + CFS, CFS and CNM. (**C**) First-order kinetic model applied for MLX release from MLX raw material, multiparticulate colonic systems CNM + CFS, CFS and CNM.

**Figure 7 pharmaceutics-14-01504-f007:**
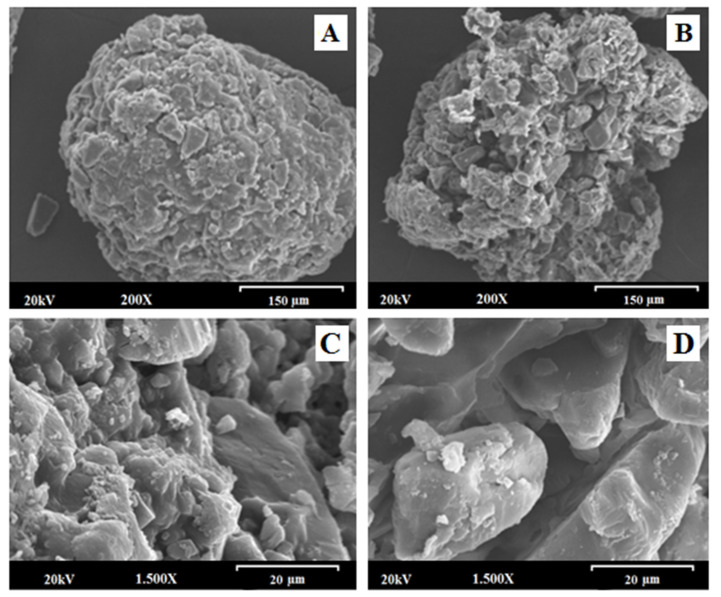
Scanning electron micrographs of CNM multiparticulate system and its blank formulation at a magnification of 200× and the scale bar is equal to 150 µm: (**A**) CNM multiparticulate system; (**B**) blank CNM formulation; Scanning electron micrographs of CNM multiparticulate system and its blank formulation at a magnification of 1500× and the scale bar is equal to 20 µm: (**C**) CNM formulation and (**D**) blank CNM formulation.

**Figure 8 pharmaceutics-14-01504-f008:**
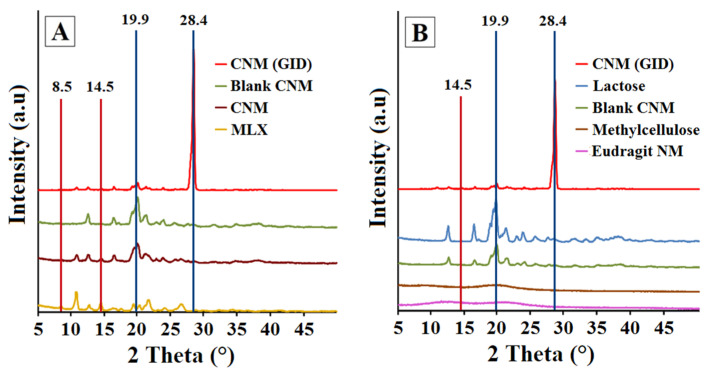
(**A**) Comparison between powder X-ray diffraction (XRD) and grazing X-ray diffraction (GID) patterns of CNM (GID); CNM (XRPD); blank CNM (XRPD) and pure MLX (XRPD). (**B**) Comparison between powder X-ray diffraction (XRD) and grazing X-ray diffraction (GID) patterns of CNM (GID); Lactose (XRPD); blank CNM (XRPD); Metolose^®^ (XRPD) and Eudragit^®^ NM (XRPD).

**Table 1 pharmaceutics-14-01504-t001:** The process parameters used in the dissolution studies.

Process Parameters	Setting
0.50–1.41 mm granule size	Dissolution rate at pH 6.8, apparatus I.
0.85–1.00 mm granule size	Dissolution rate at pH 1.2, 6.8 and 7.4, apparatus I.
0.35–1.50 mm granule size	Dissolution rate at pH 6.8 and 7.4, apparatus I.

**Table 2 pharmaceutics-14-01504-t002:** Korsmeyer-Peppas, Higuchi, and first-order kinetic models applied for MLX release from MLX raw material and multiparticulate colonic systems CNM + CFS, CFS and CNM.

Formulations	Korsmeyer-Peppas		Higuchi		First-Order	
	n	r^2^	n	r^2^	n	r^2^
FS	0.8287	0.9799	1.8851	0.9072	−0.0009	0.9023
NM + FS	1.2865	0.944	1.8552	0.9966	−0.0008	0.9943
NM	0.9818	0.9767	2.2698	0.9931	−0.0012	0.9989
CNM + CFS	0.6754	0.9337	3.3662	0.9906	−0.0023	0.9706
CFS	0.3425	0.8971	2.6235	0.8756	−0.0021	0.8412
MLX	0.6073	0.9603	4.0304	0.946	−0.0037	0.935
CNM	0.8114	0.9871	5.1589	0.9947	−0.0056	0.9995

**Table 3 pharmaceutics-14-01504-t003:** Korsmeyer–Peppas, Higuchi, and first-order kinetic models applied for MLX release from MLX raw material and multiparticulate colonic systems CNM + CFS, CFS and CNM at pH 6.8.

Formulations	Korsmeyer-Peppas		Higuchi		First-Order	
	n	r^2^	n	r^2^	n	r^2^
CNM + CFS	1.1911	0.9573	5.8765	0.9985	−0.0082	0.868
CFS	0.9702	0.9871	4.2177	0.8693	−0.0097	0.9824
MLX	0.7534	0.9464	4.1776	0.9135	−0.0058	0.9725
CNM	0.96	0.9956	5.6754	0.9884	−0.0069	0.9843

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
