# Peer review of "Multiparticulate Systems of Meloxicam for Colonic Administration in Cancer or Autoimmune Diseases"

_pharmaceutics, 2022, doi:10.3390/pharmaceutics14071504_

Round 1

Reviewer 1 Report

The manuscript “Multiparticulate systems of meloxicam for colonic administration in cancer or autoimmune diseases” by Eva Navarro-Ruíz et.al., summarizes a study to develop new colonic release systems of BCS class II drug meloxicam (MLX) a non-steroidal anti-inflammatory drug (NSAIDs) as a pH and time dependent vehicle for cancer or autoimmune diseases. The concept and idea of the work is good, but I would request the authors to revisit their manuscript and readdress all their data collected as tables and make visually appealing and easy to understand graphs and  figures. A lot has been presented in text, but the main idea of it is missing as readers do not have time to spend hours in a manuscript. And in such cases, the manuscript loses its scientific point, despite of the work being good like yours. I will proceed with a decision once changes are addressed. 

However, Kindly revise as below:

1         Line 56,The colon has a higher pH than the rest of the GIT …..”Kindly add the pH ranges, wherever a part of the anatomical system is mentioned, it’ll be easy to understand.

2         Line 63, “……pH of the stomach and proximal small intestine, but dissolve at the pH of the terminal ileum and colon..”. Kindly add the pH ranges. Also add a figure or a bar chart mentioning the pH ranges of the different sections of the GI tract. Surrounding that bar chart add the polymers and their drug release pH value and time of release.

3         Kindly add a pictorial representation of the study of the work, if possible

4         Line 247, Figure 1 to me is not acceptable, kindly use a large font while making images. The scale too is really tiny. This needs to be -redone. There are tons of manuscripts with similar research like this, why would I read this manuscript if there is nothing well presented.

5         Line 266, Figure 2 This study was done for 6hrs? Can you explain the reason why only for 6hrs? Release profile of all is within 30 %, within 30-45 mins. At 6hrs,400 mins MLX release profile is 65-70%, and CNM is helping to improve it to nearly 70 %. MLX is crashing with other polymer vehicles. In your study polymer vehicles are used to do this study. But ideally lipid-based formulations are used to promote intestinal absorption and bioavailability of the drug.

6         Line 325, Figure 3, CFS profile is good. If the diameter is presented for the granules, where are the pictures/images of the granules itself and how the diameter was measured?

7         Line 393, Figure 4 kindly use a large font while making images.

8         Line 420, Figure 5 “…homogeneity of the multiparticulate colon systems; especially in CNM with four successive granulations…”. How can a SEM image confirm homogeneity? Kindly add the particle size.

9         Line 456, Figure 6 where are the 2-theta values in the figure? Figures should be self-explanatory; no readers have time to scroll up and down of the manuscript. The way figures are presented adds a lot of attention to readers.

10     Kindly enlarge the texts, and make the figures more presentable, I could barely see with my 20/20 vision.  

11     Kindly address these few points before discussion of each section? Each of the below listed questions should be the way to address any result or discussion. This way the readers can understand the propose or motto of doing that review rather than presenting your flowcharts.

a.       What is the purpose or goal of this manuscript, section, sub-section?

b.       What new information does the result give?

c.       What methods were used to evaluate?

d.       What references were used to support the results?

Kindly re-address each section with the above questions. 

12     References. Kindly elaborate more on the sub-sections with references. Appropriate references are required to support the claim and results of any review.

13     Add pictures of the polymers, how was the microscopy done, dissolution images.

Author Response

Response to Reviewer 1 Comments

First of all, we would like to thank this reviewer for his/her comments and his/her detailed review of the manuscript.

We agree with the suggestions of this reviewer that help to clarify the manuscript.

So, we have taken into account, point by point, all reviewer suggestions to rewrite the manuscript and in order to increase the quality of our work we have performed new experimentation for more than two weeks. We hope the reviewer would appreciate all the new studies that we had to carry out.

Please see attached file with all modified figures.

Point 1           Line 56, “The colon has a higher pH than the rest of the GIT …..”Kindly add the pH ranges, wherever a part of the anatomical system is mentioned, it’ll be easy to understand.

Response 1: We have followed this suggestion of the reviewer, in order to clarify the text, we have included the pH ranges in the Introduction section (see page 2 line 54; page 2 lines 62-63; page 2 line 84).

Point 2           Line 63, “……pH of the stomach and proximal small intestine, but dissolve at the pH of the terminal ileum and colon..”. Kindly add the pH ranges. Also add a figure or a bar chart mentioning the pH ranges of the different sections of the GI tract. Surrounding that bar chart add the polymers and their drug release pH value and time of release.

Response 2: According with the referee comment we have included the pH ranges in the Introduction section (see page 2 line 54; page 2 lines 62-63; page 2 line 84). Also, we have added a figure of the GI tract with the pH ranges, the polymers (Eudragit®FS, Eudragit®NM and Metolose®), pH range of drug release (see Figure 1, Figure 1 in the corrected manuscript).

In addition, we include in the Introduction section a brief description of the characteristics of Eudragit® NM, Eudragit® FS and combination of Eudragit® FS/NM for colon-specific drug delivery (see page2 lines 63-73).

Generally, the dissolution of polymers, like Eudragit®FS and Eudragit®NM take place at terminal ileum or at ileocaecal junction when the pH exceeds 7.0 and in a short span of time drug enters into large intestine (1). Thus, formulations could have completed transit through small bowel at 6 h (2), when pH value achieves more than 7.0. The use of granules composed of different types of polymers, such as Eudragit® NM, Eudragit® FS or combinations of Eudragit® FS/NM, will allow us to select the most suitable dissolution profile for our drug when the pH value is close to 7.0.

Figure 1: Time dependent polymer, pH sensitive polymer and cellulose polymer behavior in the gastrointestinal tract.

REFERENCES

(1) Chourasia MK, Jain SK. Polysaccharides for colon targeted drug delivery. Drug Deliv. 2004 Mar-Apr;11(2):129-48. doi: 10.1080/10717540490280778. PMID: 15200012.

(2) Maurer AH. Gastrointestinal Motility, Part 2: Small-Bowel and Colon Transit. J Nucl Med Technol. 2016 Mar;44(1):12-8. doi: 10.2967/jnumed.113.134551. PMID: 26940448.

Point 3           Kindly add a pictorial representation of the study of the work, if possible

Response 3:

We have followed this suggestion of the reviewer and we have included a pictorial representation of the study of the work in the Materials and Methods section (see Figure 2, Figure 2 in the corrected manuscript).

This granulation process allows obtaining less porous formulations, compared to other processes such as lyophilization (1) (see page 4, lines 154-155).

Figure 2: Preparation procedure for MLX multiparticulate colonic systems and their characterization.

REFERENCES

(1) Li J, Lee IW, Shin GH, Chen X, Park HJ. Curcumin-Eudragit® E PO solid dispersion: A simple and potent method to solve the problems of curcumin. Eur J Pharm Biopharm. 2015 Aug;94:322-32. doi: 10.1016/j.ejpb.2015.06.002. Epub 2015 Jun 12. Erratum in: Eur J Pharm Biopharm. 2016 Jan;98:122. PMID: 26073546.

Point 4           Line 247, Figure 1 to me is not acceptable, kindly use a large font while making images. The scale too is really tiny. This needs to be -redone. There are tons of manuscripts with similar research like this, why would I read this manuscript if there is nothing well presented.

Response 4: We agree with the referee, therefore, we have redone Figure 1 (see Figure 3, Figure 3 in the revised manuscript).

Figure 3. Solubility study of MLX raw material, and multiparticulate colonic systems: FS, NM, FS+NM, CFS, CNM and CNM + CFS in phosphate buffer (pH 6.8).

Regarding the solubility experiments, we would like to remark that the selection of different polymers like Metolose® and different Eudragit® polymers NM or FS could affect the properties of drug solubility, dissolution performance, thus affecting its colonic bioavailability (1).

Important results have been found at pH 6.8 with multiparticulate colon systems of 0.50-1.41 mm diameter size:

- The solubility of MLX raw material at this pH is 249.02 ± 10.44 µg/ml.

-The solubility coefficient of meloxicam (MLX) is only improved in the formulation with Eudragit® FS and 5% of Metolose®, with a 1.09-fold increase. In all the other multiparticulate systems it is decreased compared to the pure MLX.

-The FS formulation is the one with the lowest solubility (99.90 ± 1.39 µg/ml).

-CFS and CNM multiparticulate systems with Metolose® showed a significant increase (p < 0.05) in solubility, with 2.71 and 1.38-fold increases for CFS and CNM, respectively, compared to FS and NM.

- Meanwhile, in CNM+CFS formulation Metolose® did not have any influence over MLX solubility.

MM Patel et al. (2) reported that this increase in the solubility of the MLX at this pH (small intestinal pH), which leads to increase in absorption from the small intestinal region.

We have highlighted these interesting results in the text of the Results and Discussion section and in the Conclusion section (see page 8 lines 289-295 and page 17 lines 578-579, respectively).

REFERENCES

(1) Lu, J., Obara, S., Liu, F., Fu, W., Zhang, W., & Kikuchi, S. (2018). Melt extrusion for a high melting point compound with improved solubility and sustained release. AAPS PharmSciTech, 19(1), 358-370

(2) Mayur M. Patel & Avani F. Amin (2011) Formulation and development of release modulated colon targeted system of meloxicam for potential application in the prophylaxis of colorectal cancer, Drug Delivery, 18:4, 281-293, DOI: 10.3109/10717544.2010.538447

Point 5           Line 266, Figure 2 This study was done for 6hrs? Can you explain the reason why only for 6hrs? Release profile of all is within 30 %, within 30-45 mins. At 6hrs,400 mins MLX release profile is 65-70%, and CNM is helping to improve it to nearly 70 %. MLX is crashing with other polymer vehicles. In your study polymer vehicles are used to do this study. But ideally lipid-based formulations are used to promote intestinal absorption and bioavailability of the drug.

Response 5: We would like to thank the referee for this comment. We have repeated these dissolution rate studies for 24 h under the same starting conditions. These studies showed similar dissolution profiles and have allowed us to include the 7 and 8 h data (see Figure 4, Figure 4 in the corrected manuscript). We believe that not including the 24-hour data in Figure 4 improves the resolution by showing the time axis for only 8 hours.

Regarding lipid-based formulations, we agree with the referee that these lipid components could promote the absorption of the drug and, therefore, its bioavailability. This increase in absorption has been attributed to the lymphatic absorption of different nanoemulsions or nanocarriers (1-3), but with larger particles this lymphatic absorption is not favored (4).

However, in recent years, there have been different works that show how the Eudragit® NM polymer improves the absorption of different drugs (5), which has been related in many cases to an increase in solubility at pH 7. We propose to include the following result:

“As it can be seen in Figure 4, CNM granules achieved the maximum percentage re-leased of MLX at 8h (88.77 ± 3.46%), yielding more than the reference active ingredient, the pure MLX (78.10 ± 3.01% at the same time), followed by the CFS, with 66.34 ± 2.12% and CNM + CFS with 61.86 ± 2.60% at 8 h. Different studies have shown how the improvement in drug solubility in a pH zone (6.5-7) increases drug solubility at the colon site (6,7)” (see pages 9-10, lines 334-339).

Figure 4. Release profiles at pH 6.8 for MLX raw material and multiparticulate colon systems of 0.50-1.41 mm diameter size: FS, NM, NM + FS, CFS, CNM and CNM + CFS.

REFERENCES

(1) Porter, C. J., & Charman, W. N. (2001). Intestinal lymphatic drug transport: an update. Advanced drug delivery reviews, 50(1-2), 61-80.

(2) Fricker, G., Kromp, T., Wendel, A., Blume, A., Zirkel, J., Rebmann, H., ... & Müller-Goymann, C. (2010). Phospholipids and lipid-based formulations in oral drug delivery. Pharmaceutical research, 27(8), 1469-1486.

(3) Plaza-Oliver, M., Santander-Ortega, M. J., & Lozano, M. V. (2021). Current approaches in lipid-based nanocarriers for oral drug delivery. Drug delivery and translational research, 11(2), 471-497.

4) Rani, S., Rana, R., Saraogi, G. K., Kumar, V., & Gupta, U. (2019). Self-emulsifying oral lipid drug delivery systems: advances and challenges. AAPS PharmSciTech, 20(3), 1-12.

(5) Jennotte, Olivier & Koch, Nathan & Lechanteur, Anna & Evrard, Brigitte. (2022). Development of amorphous solid dispersions of cannabidiol: Influence of the carrier, the hot-melt extrusion parameters and the use of a crystallization inhibitor. Journal of Drug Delivery Science and Technology. 71. 103372. 10.1016/j.jddst.2022.103372.

(6) Zhang F. Melt-Extruded Eudragit® FS-Based Granules for Colonic Drug Delivery. AAPS PharmSciTech. 2016 Feb;17(1):56-67. doi: 10.1208/s12249-015-0357-2.

(7) Mayur M. Patel & Avani F. Amin (2011) Formulation and development of release modulated colon targeted system of meloxicam for potential application in the prophylaxis of colorectal cancer, Drug Delivery, 18:4, 281-293, DOI: 10.3109/10717544.2010.538447.

Point 6           Line 325, Figure 3, CFS profile is good. If the diameter is presented for the granules, where are the pictures/images of the granules itself and how the diameter was measured?

Response 6: Figure 3 (Figure 5 in the revised manuscript) showed that the granulate that released the most MLX at initial times was CFS, achieving a 44.02 ± 0.55 % and 75.90 ± 1.44 % at 30 min and 60 min respectively, but also the CFS profile was uncontrolled and possessed a great burst effect. Following oral administration, these systems are subjected to challenges from the harsh and hostile environment of the GI tract before reaching the colon. For instance, the unwanted burst drug release in the stomach and the small intestine before reaching the colon is an intrinsic limitation of colon-targeted drug delivery systems (1) (see page 11, lines 394-400). We chosen CNM formulation because it presented a controlled sustained release at this pH, moreover, in the following dissolution study at 3 pHs (1.2; 6.8 and 7.4) the best formulation was also CNM. Therefore, we performed the SEM studies only with this formulation.

However during the revision of our manuscript we have carried out new SEM studies with the following multiparticulate systems: CFS, CFS+CNM and CNM formulations (see Figure 6, formulations after their preparation).

Regarding how the diameter was measured, we selected the particle size for the different dissolution studies by sieving. The sieving method is a conventional and quantitative method for particle size determination. It, however, requires a high amount of the sample under investigation for each measurement (2) (page 4 lines 157-161).  

Figure 5. Release profiles of multiparticulate colonic systems of 0.35-1.50 mm diameter size (CNM, CFS and CNM + CFS) and pure MLX in pH 6.8 medium.

Figure 6. Scanning electron micrographs of (A) CFS, (B) CNM + CFS and (C) CNM multiparticulate system. Original magnification is 200 × and the scale bar is equal to 200 µm.

REFERENCES

(1) Naeem, M., Awan, U.A., Subhan, F. et al. Advances in colon-targeted nano-drug delivery systems: challenges and solutions. Arch. Pharm. Res. 43, 153–169 (2020). https://doi.org/10.1007/s12272-020-01219-0

(2) Maruyama S, Ando S, Yonemochi E. Application of void forming index (VFI): Detection of the effect of physical properties of dry powder inhaler formulations on powder cohesion. Int J Pharm. 2020 Oct 15;588:119766. doi: 10.1016/j.ijpharm.2020.119766. Epub 2020 Aug 13. PMID: 32800937.

Point 7           Line 393, Figure 4 kindly use a large font while making images.

Response 7: We have followed the suggestion of the reviewer and we have corrected Figure 4 (see Figure 7, Figure 6 in the corrected manuscript).

Figure 7. (A) Dissolution rate of MLX with buffer changes (pH 1.2, 6.8 and 7.4) for MLX raw material and 0.85-1.00 mm size diameter granules of CNM, CFS and CNM + CFS. (B) Higuchi kinetic model applied for MLX release from MLX raw material, multiparticulate colonic systems CNM + CFS, CFS and CNM. (C) First-order kinetic model applied for MLX release from MLX raw material, multiparticulate colonic systems CNM + CFS, CFS and CNM.

Point 8           Line 420, Figure 5 “…homogeneity of the multiparticulate colon systems; especially in CNM with four successive granulations…”. How can a SEM image confirm homogeneity? Kindly add the particle size.

Response 8: We agree with the referee, SEM image cannot confirm homogeneity of a sample, we have deleted this part. Furthermore, we have included a micrograph at a magnification of 20x (see Figure 8, Figure S1 in the revised manuscript) in order to identify the approximately particle morphology, size, and shape by SEM. Benavent C et al (1) performed an SEM analysis identifying its surface and an approximate particle size range for different solid dispersions. In Figure 8.A (CNM formulation) it is shown irregular particles with a matrix structure and a rough surface (0.51-0.24 mm) and in Figure 8.B (blank CNM formulation) similar structures are shown with a slight larger particle size (0.67-0.27 mm). According with the referee comment we have rephrased this part in the Results section (see page 14, lines 499-505).

Figure 8. Scanning electron micrographs of CNM multiparticulate system and its blank formulation, at a magnification of 20× and the scale bar is equal to 1.5 mm: (A) CNM multiparticulate system; (B) blank CNM formulation.

REFERENCES

(1) Benavent C, Torrado-Salmerón C, Torrado-Santiago S. Development of a Solid Dispersion of Nystatin with Maltodextrin as a Carrier Agent: Improvements in Antifungal Efficacy against Candida spp. Biofilm Infections. Pharmaceuticals (Basel). 2021 Apr 22;14(5):397. doi: 10.3390/ph14050397.

Point 9           Line 456, Figure 6 where are the 2-theta values in the figure? Figures should be self-explanatory; no readers have time to scroll up and down of the manuscript. The way figures are presented adds a lot of attention to readers.

Response 9: We have followed the suggestion of the reviewer and we have corrected Figure 6 including the following 2-theta values: majority peaks of the MLX at angle 2 (θ) positions of 8.5 ± 0.2° and 14.5 ± 0.2° (1) and characteristic lactose peaks at 16.4 ± 0.2° and 17.4 ± 0.2° (2) (see Figure 9, Figure 7 in the corrected manuscript).

Figure 9. (A) Comparison between powder X-ray diffraction (XRD) and grazing X-ray diffraction (GID) patterns of CNM (GID); CNM (XRPD); blank CNM (XRPD) and pure MLX (XRPD). (B) Comparison between powder X-ray diffraction (XRD) and grazing X-ray diffraction (GID) patterns of CNM (GID); Lactose (XRPD); blank CNM (XRPD); Metolose® (XRPD) and Eudragit®NM (XRPD).

REFERENCES

(1) Jacon, J.; Santos, O.; Bonfilio, R.; Doriguetto, A.; de Araújo, M. Analysis of polymorphic contamination in meloxicam raw materials and its effects on the physicochemical quality of drug product. Eur. J. Pharm. Sci. 2017, 109, 347-358. doi.org/10.1016/j.ejps.2017.08.029.

(2) Abu Fara D, Rashid I, Alkhamis K, Al-Omari M, Chowdhry BZ, Badwan A. Modification of α-lactose monohydrate as a direct compression excipient using roller compaction. Drug Dev Ind Pharm. 2018 Dec;44(12):2038-2047. doi: 10.1080/03639045.2018.1508224. Epub 2018 Sep 5. PMID: 30095020.

Point 10         Kindly enlarge the texts, and make the figures more presentable, I could barely see with my 20/20 vision. 

Response 10: We have followed the suggestion of the reviewer and we have corrected all figures. Regarding the size of the text we have used the template from the Pharmaceutics, and it has a small size of text, however we have included the corrected manuscript after the response of all your comments.

Point 11       Kindly address these few points before discussion of each section? Each of the below listed questions should be the way to address any result or discussion. This way the readers can understand the propose or motto of doing that review rather than presenting your flowcharts.

  1. What is the purpose or goal of this manuscript, section, sub-section?
  2. What new information does the result give?
  3. What methods were used to evaluate?
  4. What references were used to support the results?

Kindly re-address each section with the above questions. 

Response 11: We agree with the referee. We have re-addressed each section in order to better explain its purpose, new information, methods used and main references to support results.

Point 12         References. Kindly elaborate more on the sub-sections with references. Appropriate references are required to support the claim and results of any review.

Response 12: as suggested, we have included new references that support the results observed in the different sub-sections.

Point 13         Add pictures of the polymers, how was the microscopy done, dissolution images.

Response 13: We have followed this suggestion of the reviewer, and we have included SEM micrographs to examine the surface morphology of Metolose®, Eudragit® FS and Eudragit® NM in their pure states (see Figure 10, Figure S2 in the corrected manuscript). According with the referee comment we have included this part in the Results section (see page 14, lines 506-511). In Figure 10.A methylcellulose (Metolose®) appears as irregular flake fibers, also described by other authors (1). On the other hand, Eudragit® FS and NM displayed a very smooth film surface, with some scales of different thickness, this smooth surface has been previously reported in formulations with Eudragit® (2).

Figure 10. Scanning electron micrographs of polymers at a magnification of 200× and the scale bar is equal to 200 µm: (A) Metolose®; (B) Eudragit®FS and (C) Eudragit ®NM.

Regarding the dissolution test, we have performed SEM surface images which shall give information about changes in the surface morphology during dissolution (1), simulating the gastro-intestinal conditions: pH 1.2 (gastric pH), pH 6.8 (small intestine pH) and pH 7.4 (colon pH). We show the referee the SEM images of CFS, CNM+CFS and CNM systems at initial time (after preparation, see Figure 6) and multiparticulate systems during the dissolution test after 2 h at pH 1.2, 3 h at pH 6.8 and 3 h at pH 7.4 (2, 5 and 8 h) that were withdrawn from the dissolution medium and lyophilized (see Figure 11, Figure S3 in the corrected manuscript).

Figure 11 shows scanning electron micrographs of the surfaces of multiparticulate systems CFS, CNM+CFS and CNM, after 2 h at pH 1.2; 3 h at pH 6.8 and 3 h at pH 7.4. After 2 h in simulated gastric fluid (pH 1.2) the surface structure of all three formulation remained very similar to the initial time (t0), presenting a very rough surface. However after being in an alkaline medium (pH 6.8), the three of the multiparticulate systems showed a more porous structure. Similar studies were carried out by several authors (1,3,4). Therefore SEM images (pH 6.8) let assume that the dissolution happens through pores that are generated by a combination of drug diffusion (1) and polymers swelling / eroding processes (3). Meanwhile, after being 3 h in pH 7.4 CFS and CNM+CFS systems showed a smoother structure, this coul be related to a rapid polymer disintegration. In contrast, the CNM formulation still had a rough surface, maintaining its structure. This result may be related to its more controlled dissolution rate. Similar results were obtained by Mehta R. et al (4) in Eudragit® S-100 coated naproxen matrix tablets for colon-targeted drug delivery system.

Figure 11. Scanning electron micrographs of CFF, CNM + CFS and CNM multiparticulate systems, after 2 h at pH 1.2; 3h at pH 6.8 and 3 h at pH 7.4. Original magnification is 1500x and the scale bar is equal to 30µm.

According with the referee comment we have included this part in the Results and Discussion section (see page 14-15, lines 512-525).

REFERENCES

(1) Güres S, Kleinebudde P. Dissolution from solid lipid extrudates containing release modifiers. Int J Pharm. 2011 Jun 30;412(1-2):77-84. doi: 10.1016/j.ijpharm.2011.04.010. Epub 2011 Apr 14. PMID: 21515350.

(2) Mustafa, W.W., Fletcher, J., Khoder, M. et al. Solid Dispersions of Gefitinib Prepared by Spray Drying with Improved Mucoadhesive and Drug Dissolution Properties. AAPS PharmSciTech 23, 48 (2022). https://doi.org/10.1208/s12249-021-02187-4.

(3) Guarnizo-Herrero V, Torrado-Salmerón C, Torres Pabón NS, Torrado Durán G, Morales J, Torrado-Santiago S. Study of Different Chitosan/Sodium Carboxymethyl Cellulose Proportions in the Development of Polyelectrolyte Complexes for the Sustained Release of Clarithromycin from Matrix Tablets. Polymers (Basel). 2021 Aug 21;13(16):2813. doi: 10.3390/polym13162813. PMID: 34451351; PMCID: PMC8400629.

(4) Mehta R, Chawla A, Sharma P, Pawar P. Formulation and in vitro evaluation of Eudragit S-100 coated naproxen matrix tablets for colon-targeted drug delivery system. J Adv Pharm Technol Res. 2013 Jan;4(1):31-41. doi: 10.4103/2231-4040.107498. PMID: 23662280; PMCID: PMC3645356.

CORRECTED MANUSCRIPT: Multiparticulate systems of meloxicam for colonic administration in cancer or autoimmune diseases

Abstract: The aim of this research is the development of new colonic release systems of meloxicam (MLX) a non-steroidal anti-inflammatory drug (NSAIDs) with pH and time dependent vehicles for cancer or autoimmune diseases. The colon has a higher pH than the rest of the gastrointestinal tract (GIT) and this can be used as a modified release strategy. Eudragit® polymers are the most widely used synthetic products in the design of colonic release formulations because they might offer mucoadhesiveness and pH-dependent release. Colonic de-livery systems produced with pH-dependent and permeable polymers (FS-30D) or with pH- independent and low permeability polymers (NM-30D), must dissolve at a pH range of 6.0-7.0 to delay the release of the drug and prevent degradation in the GIT, before reaching the colon. The conditions prepared simulating a gastrointestinal transit showed the CNM multi-particulate system, composed of Eudragit® NM and cellulose, as the best release option for MLX with a more sustained release with respect to the other formulations. CNM formulation followed Higuchi and First-order release kinetics, thus MLX release was controlled by a combination of diffusion and polymers swelling / eroding processes.

Keywords: multiparticulate system; autoimmune disease; cancer, colonic administration, meloxicam.

  1. Introduction

Colonic delivery systems aim to release the active substance in the last portion of the small intestine, without releasing it in the upper gastrointestinal tract (GIT), so it is free in the first portion of the large intestine, the ascending colon, which is a window of absorption for many high molecular weight substances. Successful colonic release requires the drug to reach the ascending colon in a precise time. In addition, the loss of active ingredient should be minimized because of the enzymatic activity of the ileum or the sequestration of the drug by the stool already compacted in the distal colon. Therefore, the objectives of an oral dosage form prepared for a colonic absorption are aimed to protect the drug throughout its gastrointestinal transit to the colon, to standardize the residence time at the colonic level, to guarantee system recognition by the colonic mucosa and to ensure a specific release zone. Hence, for a colonic release it is important to consider the physiological properties of the colon. In general, the GIT undergoes changes in motility, content, enzyme activity and pH from the stomach to the intestine [1]. To optimize colonic release systems, several possibilities have been studied, and we have highlighted the pH and time-dependent release systems [2].

The selective release systems at the colonic level must delay the release, which constitute an interesting alternative for the administration of some drugs in a less hostile environment than the stomach or small intestine with a high enzymatic activity [3-5]. It is also important the long residence times in this area of the intestine, increasing the systemic absorption of some drugs and achieving a local effect [6-9].

The colon has a higher pH (6.8-7.5) than the rest of the GIT (1.2-6.8) [10-12] and this can be used as a modified release strategy. In previous works [13-14] some pH-dependent formulations were prepared with polymers such as cellulose acetate phthalate (CAP), hydroxypropyl methyl cellulose pthalates (HPMCP) 50 and 55, and copolymers of methacrylic acid and methyl-methacrylate (for example, Eudragit® S100, Eudragit® L, Eudragit® FS and Eudragit® P4135F). Eudragit® polymers are the most widely used synthetic products in the design of colonic release formulations because they offer mucoadhesiveness and pH-dependent release [15-18]. The ideal polymer should be able to withstand the low pH of the stomach (pH 1-3) and proximal small intestine (pH 4.5-6.0), but dissolve at the pH of the terminal ileum and colon (pH 6.5-7.5) (see Figure 1). As a result, colonic delivery systems produced with pH dependent polymers must dissolve at a pH of 6.0-7.0 to delay the release of the drug and prevent degradation in the GIT before reaching the colon. Generally, the dissolution of polymers, like Eudragit®FS and Eudragit®NM, take place at terminal ileum or at ileocaecal junction when the pH exceeds 7.0 and in a short span of time drug enters into large intestine [19]. Thus, formulations could have completed transit through small bowel at 6 h [20] when pH value achieves more than 7.0. Eudragit® NM, Eudragit® FS and Eudragit® FS/NM combinations have been used for colonic drug delivery. The high flexibility of Eudragit® NM favors compactness and better maintains the multiparticle structure in colonic delivery systems [21].

Some studies [10] of delayed and uniform release used Eudracol®, system which is based on coating the pellet with Eudragit® RL/RS and Eudragit® FS30D, giving a specific colonic release in a pH and time dependent way. To improve specificity for the colon, Naeem et al. [11] manufactured pH and time-dependent budesonide nanoparticles for the treatment of ulcerative colitis. These nanoparticles were prepared with Eudragit® FS30D and Eudragit® RS100. Eudragit® FS30D is a pH dependent polymer that dissolves in environments above 7.0, while Eudragit® RS100 is a time dependent controlled release polymer with low permeability. By combining these two polymers, premature release in the upper gastrointestinal tract is effectively minimized, achieving the desired colonic release.

The pH of the colon, which varies from 6.5 to 7.5, in normal conditions, is very sensitive to alterations by diet, diseases, water intake, and microbial metabolism [12]. For example, patients with ulcerative colitis have a more acidic colonic pH compared to healthy individuals. This would give incomplete release of enteric-coated systems [22]. The peristaltism of the colonic segment is slow, and the content remains in the colon for a long time, therefore, the development of time-dependent colonic formulations is also interesting [23]. Time-dependent colonic release systems guide the release of the drug by the arrival times of the food after oral administration. In general, gastric emptying time is 15-180 min, while retention in the small intestine is 3-4 h. Thus, the release of time-dependent systems is usually around 5–6 h [24]. Currently, the drug is coated with insoluble coatings that are difficult to decompose and the release time is controlled by the proportion and dose of the coating material [25]. Eudragit® NM 30 D, neutral, with low permeability is another attractive option, since it can be formulated with all active ingredients regardless of the ionic charge they have and in case highly flexible films are required.

Figure 1. Time dependent polymer, pH sensitive polymer and cellulose polymer behavior in the gastrointestinal tract.

MLX is BCS class II drug (meloxicam, MLX) (4-hydroxy-2-methyl-N-(5-methyl-2-thiazolyl)-2H-1,2-benzothiazine-3-carboxamide) [26]. Moreover, it is a non-steroidal anti-inflammatory drug (NSAIDs) that is distinguished by inhibiting COX-2 to a greater degree than COX-1 in both its cyclooxygenase and peroxidase activity. Because MLX is well absorbed in the colonic region, its properties against colon cancer and autoimmune anti-inflammatory diseases, it is interesting to study new colonic formulations of MLX [27-28]. Nevertheless, fewer studies have considered the development of colonic formulations using multiparticulate systems like pellets or granules of MLX. In fact, MLX tablets containing Eudragit® FS 30D, as outer coating layer, were prepared for colon targeting [25]. The decline in pH from the end of the small intestine to the colon can also result in problems, lengthy lag times at the ileo-cecal junction or rapid transit through the ascending colon which can also result in poor site-specificity for enteric-coated single-unit formulations [3]. In addition, multiparticle systems hold great potential for enhancing drug targeting as well as drug uptake. Therefore, as shown by several authors, various formulations with particle size reduction may be beneficial for colon-targeted drug delivery [29], such as, MLX pellets with Eudragit® FS [30] and MLX polymeric micelles based in amphiphilic chitosan derivatives [31]. Pellets production process is very difficult to control, and also very costly, meanwhile polymeric micelles present temporal stability and additional concerns might be linked to their drug loading capacity. However, development of MLX granulates, could be a good candidate for colon targeting, because of their simple manufacturing process, good stability, homogeneous drug load and could provide a site-specific drug delivery [32].

The prevalence of colonic diseases has increased worldwide, demanding the effectiveness of local treatments for more effective and safer therapies. Of all colonic diseases, colorectal cancer (CRC) causes the highest number of cancer deaths in Europe (with more than 200,000 deaths annually and is the third most diagnosed cancer in the world [33]. The incidence of inflammatory bowel disease (IBD) is also growing alarmingly in areas such as Asia, where there was previously low incidence. Consequently, the need for effective treatment for colonic diseases is one of the world's public health problems.

This work explores MLX-colonic release systems that would be a good proposal for the treatment of colonic diseases, thanks to its anti-inflammatory effect. Its usefulness in chemoprevention, chemo-suppression and UV protection has recently been studied [27-28]. The purpose of this research is the development of new colonic release systems of MLX with pH and time dependent vehicles, avoiding pH degradation across the GIT, for cancer or autoimmune diseases.

  1. Materials and Methods

2.1. Materials

Meloxicam (MLX, Fagron Ibérica, Barcelona, Spain); Methylcellulose (Metolose® 90 SH 100, Shin-Etsu Chemical Co., Ltd., Tokyo, Japan); aqueous dispersion of a copolymer of ethyl acrylate and methyl methacrylate (Eudragit® FS 30D and Eudragit® NM 30D, Evonik Röhm GmbH, Germany); a-lactose monohydrate (Fagron Ibérica, Barcelona, Spain). All other chemicals were of analytical grade.

2.2. Preparation of Formulations

We previously described the development of these colonic granulates [32], in the present study we reformulate them but selecting different particle sizes. Briefly, the studied MLX was granulated with lactose and two types of Eudragit®: FS and NM; some granules containing 5% of cellulose (Metolose® 90 SH 100). The granules are made in four successive granulations, leaving a day of drying between each one, until approximately 30% of Eudragit® is added to the different formulations. A Blank formulation, with the four successive granulations, without MLX, containing lactose, Metolose® and Eudragit® NM was manufactured for their studying in either scanning electron microscopy (SEM), X-ray powder diffractometry (XRPD) or grazing incidence X-ray diffraction (GID), respectively. This granulation process allows obtaining less porous formulations, compared to other processes such as lyophilization [34]. A summary of this process is shown in Figure 2.

Finally, different diameters were selected by sieving to determine their in vitro behavior at different pH buffers, in dissolution studies (apparatus I), which are presented in Table 1. The sieving method is a conventional and quantitative method for particle size determination. It, however, requires a high amount of the sample under investigation for each measurement [35].

After the dissolution rate data obtained for MLX, the most suitable formulation would be selected and prepared by wet granulation of the MLX with 30 % Eudragit® FS or NM alone, or with a combination of both polymers NM+FS (1:1), hereinafter NM+FS, either adding a 5 % cellulose (Metolose®) obtaining CFS, CNM or CNM + CFS (1:1), hereinafter CNM + CFS. Subsequently, the physicochemical characterization of the best formulation in the in vitro tests was carried out using SEM, XRPD or GID analytical techniques.

Figure 2: Preparation procedure for MLX multiparticulate colonic systems and their characterization.

Table 1. The process parameters used in the dissolution studies.

Process parameters

Setting

0.50-1.41 mm granules size

Dissolution rate at pH 6.8, apparatus I.

0.85-1.00 mm granules size

Dissolution rate at pH 1.2, 6.8 and 7.4, apparatus I.

0.35-1.50 mm granules size

Dissolution rate at pH 6.8 and 7.4, apparatus I.

2.3. Solubility Study

Solubility tests of the granules (FS, NM, NM+FS, CFS, CNM, CNM + CFS) were carried out with 3 mg of MLX or its equivalent amount in 0.5-1.41 mm diameter size granules, in 2 mL of pH 6.8 buffer in a thermostatic bath at 37 ± 1°C stirring for 2 days. Samples were filtered and diluted 1/50 with a 6.8 buffer to be quantified. The cumulative amount of MLX released from the formulations was determined at 362 nm with a UV-VIS JASCO V-730 spectrophotometer (Jasco® International Co. Ltd.; Tokyo, Japan), with the following calibration curve: y = 0.0508 х (µg/mL) + 0.0105 (r2 = 0.9995) across a range of 2–15 µg/mL. Each determination at each time was performed in triplicate, and the error bars on the graphs represent the standard deviation.

2.4. In Vitro Release Profile Study 

To evaluate MLX in the different granulations, a quick and simple analytical method has been developed: an UV spectrophotometry technique. Different wavelengths were selected to carry out the normal MLX spectrophotometric test according to the pH study, taking the wavelength at which, the MLX presented a maximum absorption in the different pH medium. At pH 7.4, the cumulative amount of MLX released from the formulations was determined at 361 nm with a UV-VIS JASCO V-730 spectrophotometer (Jasco® International Co. Ltd.; Tokyo, Japan), with the following calibration curve: y = 0.0512 х (µg/mL) + 0.0141 (r2 = 0.9995) across a range of 2–15 µg/mL. At pH 6.8, the MLX released from the formulations was determined at 362 nm by the spectrophotometric method described in the solubility study. Then at pH 1.2, the cumulative amount of MLX released from the formulations was determined at 346 nm with a UV-VIS JASCO V-730 spectrophotometer (Jasco® International Co. Ltd.; Tokyo, Japan), with the following calibration curve: y = 0.0377 х (µg/mL) + 0.0368 (r2 = 0.9999) across a range of 1–15 µg/mL. The determination at each time point was performed in triplicate, and the error bars on the graphs represented the standard deviation. Dissolution studies were performed using the United States Pharmacopeia (USP) basket method (apparatus 1) in ERWEKA® DT 80 (ERWEKA GmbH; Langen, Germany) dissolution equipment. The temperature was maintained at 37.0 ± 0.1 °C throughout the dissolution study.

Dissolution rate at pH 6.8: The first dissolution rate test (apparatus I) was carried out with the 0.50-1.41 mm granules, which contained 10 mg of MLX. The vessels contained 900 mL of pH 6.8 phosphate buffer, at 37 ± 0.1°C, with a rotation speed of 50 rpm. Taking samples at 5, 15, 30, 45, 60, 120, 180, 240, 300, 360 and 390 min. The samples were filtered through 0.45 µm (Millipore®) and analyzed directly. 

The dissolution rate of the granules of 0.350-1.50 mm were studied at pH 6.8, 37.0°C ± 0.1 and 50 rpm. The test lasts 8 h, the samples were taken at 5, 15, 30, 45, 60 min and then every hour until 8 h. Samples were filtered through 0.45 µm and analyzed at 362 nm.

The dissolution test in pH 7.4-phosphate buffer, with granules of 0.350-1.50 mm, at 50 rpm and 37.0°C±0.1. Reading the samples at 5, 15, 30, 45 and 60 min and then every hour until 8 h, then filtering and analyzing at 361 nm wavelength.

The last dissolution test was carried out at pH 1.2, 6.8 and 7.4 with buffer changes: A test was carried out starting from 900 mL of pH 1.2 in the apparatus I, at 37 ± 0.1°C and with the granules of size 0.85-1.00 mm, at a speed of 50 rpm. It was made with CFS, CNM and CNM + CFS granules. Samples were taken at 1 h and two hours later, we removed and changed the medium trying not to lose any granules, filling up to 900 mL with pH 6.8 buffer. Samples were then taken every half hour, at 2.5, 3, 3.5, 4, 4.5 and 5 h, to change the medium at 5 h, we changed the medium to pH 7.4, taking samples at 5.5, 6, 6.5, 7, 7.5 h, and finally at 24 h. Samples were filtered at 0.45 µm.

To investigate the effect either of different particulate diameters or of diverse polymers on the release of MLX more precisely, the results were analyzed according to the Korsmeyer-Peppas Equation for Mt/M < 0.6, which can be expressed as the following Equation:

Mt/M= Kd·tn                      (1)

where Mt/M is the fractional drug released at time t (h) from the total amount released M. Kd (min-1) is the kinetic dissolution constant, and n is a diffusional exponent characteristic of the release as a function of time t. For drug release from tablets, which contains hydrophilic polymers, the exponent n is the diffusional constant that characterizes the drug release transport. When n = 0.5, a Fickian diffusion process was observed, where drug diffusion through the polymeric matrix is the dominant release mechanism.

When n values are between 0.5 < n < 1, drug diffusion occurs via anomalous transport (non-Fickian). In anomalous diffusion, it is assumed that the mechanism of CL release is a combination of swelling, erosion, and diffusion. When n = 1, Case II transport or zero order release kinetics could be observed. An n value of about 1.0 indicates that polymer relaxation, polymer dissolution, or tablet erosion are the dominant mechanisms [36]. For the mathematical evaluations, we characterized drug release kinetics by fitting standard release to Higuchi, zero-order and first-order models [37].

Mt/M= K0·t   zero-order model       (1)

Mt/M= KH·t0.5 Higuchi model         (2)

Ln (Mt/M)= -K1·t first-order model    (3)

where Mt/Mis the fractional drug released at time t (min) from the total amount released M KH, K0, and K1 (min-1) are the kinetic dissolution constants for zero-order, Higuchi, and first-order kinetic models, respectively, which characterize release as a function of time t.

The high r2 values in the zero-order model indicate that polymer relaxation or tablet erosion is the dominant mechanism and is related to Case II transport for the Korsmeyer-Peppas Equation. The high r2 values in the Higuchi model indicate the better fit of release data for diffusion kinetics, which are related to a Fickian diffusion process in the Korsmeyer-Peppas equation. The high r2 values in the first-order model indicated swelling and erosion phenomena and could be related to non-Fickian (anomalous transport) for the Korsmeyer-Peppas equation.

2.5. Scanning Electron Microscopy (SEM)

The samples were mounted on an aluminum sample mount. After coating with a thin layer of gold-palladium, the hydrogel samples were analyzed via SEM using a Jeol®6400. All micrographs were the product of secondary electron imaging used for surface morphology identification at different magnifications and an accelerating voltage of 20 kV.

Samples of CNM, CFS and CNM + CFS multiparticulate systems during the dissolution test after 2 h at pH 1.2, 3 h at pH 6.8 and 3 h at pH 7.4 (2, 5 and 8 h) were withdrawn from the dissolution medium and lyophilized, then they were analyzed by SEM.

2.6. X-ray Powder Diffractometry (XRPD) and Grazing Incidence X-ray Diffraction (GID)

XRPD was used for characterizing crystalline materials and provides information on structures. The dust and surface sweep method were used. Approximately 20 mg of sample was dispersed on a sample holder and loaded into a Philips X'PERT® diffractometer equipment. Each sample was exposed to Cu-kα radiation, the angles of incidence being from 5° to 50°. The test conditions are 40 mV voltage and 55 mA intensity. The diffractograms obtained from the raw materials (MLX, lactose, Metolose® and Eudragit®) and from the CNM formulation and its placebo make it possible to know if the samples are crystalline or amorphous. GID method has also been used, which consists in that the angle of incidence (θ) remains fixed while the detector moves normally around the axis of the goniometer [38]. GID is a surface-sensitive technique that, allows characterization of epitaxial films with thicknesses down to a few atomic layers. All measurements were made with a Philips X'PERT® diffractometer, where the radiation was the same as for the measurements by the powder method. In this study, we have used an incidence angle (θ) of 1°. Before the measurements, the equipment was carefully aligned and calibrated. A spatial sample holder was used that could be adjusted to the height of the sample. Under these conditions, the CNM formulation was measured.

2.7. Statistical Analysis

Differences between obtained values (Mean ± SE) for the prepared formulae and the control formulae were carried out using one-way analysis of variance (ANOVA), followed by an appropriate post-hoc test in the case of the presence of a significant difference. A p-value less than 0.05 was considered a criterion for a statistically significant difference.

  1. Results and Discussion

3.1. Solubility Studies at pH 6.8 with MLX Raw Material and 0.50-1.41 mm Granules of Formulations: FS, NM, NM + FS, CFS, CNM and CNM + CFS.

The purpose of this study is to evaluate the influence of Metolose® and different Eudragit® polymers NM or FS on MLX solubility properties, which could affect its dissolution performance, thus affecting its colonic bioavailability [39].

As can be seen in Figure 3, the solubility of MLX raw material at this pH is 249.02 ± 10.44 µg/mL. The drug solubility coefficient is only slightly improved in the formulation with CFS granules (1.09-fold) compared to pure MLX. In all the other multiparticulate systems, it is decreased compared to the pure MLX. The FS formulation is the one with the lowest solubility (99.90 ± 1.39 µg/mL). CFS and CNM multiparticulate systems with Metolose® showed a significant increase (p < 0.05) in solubility, with 2.71 and 1.38-fold increases for CFS and CNM, respectively, compared to FS and NM. Metolose® seems to help granules to disintegrate, achieving higher dissolution values [40]. Meanwhile, in CNM + CFS formulation Metolose® did not have any influence over MLX solubility. Probably, Eudragit® FS together with Eudragit® NM are forming a more cohesive film coat, avoiding water absorbance of Metolose® [41].

Figure 3. Solubility study of MLX raw material, and multiparticulate colonic systems: FS, NM, NM+FS, CFS, CNM and CNM + CFS in phosphate buffer (pH 6.8).

In other studies, Weyna et al. [42] analyzed the solubility of MLX at pH 6.5 with respect to co-crystals of the active ingredient, correlating the results in vitro with in vivo test performed in rats, administering a suspension with the equivalent amount of 10 mg/kg of MLX. It was shown that samples with a greater solubility at that pH also had a higher absorption and an earlier onset of action, so it is expected that the granules with a solubility coefficient higher than MLX raw material will have a higher absorption in vivo. In addition, Patel et al. [25] reported that this increase in the solubility of the MLX at this pH (small intestinal pH), which leads to increase in absorption from the small intestinal region.

In the solubility studies, the formulations with Metolose® experienced an increase in solubility, with the exception of the CNM + CFS granulate. It was decided to verify these results by performing a dissolution rate test in a pH 6.8 medium, at 37°C for pure MLX and all formulations: FS, NM, NM + FS, CFS, CNM and CNM + CFS.

3.2. In Vitro Release Profile Study

This study is important to evaluate the influence of Metolose® and different Eudragit® polymers NM or FS on MLX dissolution performing at different pHs simulating the gastro-intestinal environment, which could affect its colonic bioavailability [39]. To investigate the effect either of different particulate sizes or of diverse polymers on the release of MLX more precisely, the results were analyzed according to the Korsmeyer-Peppas Equation. In addition, it was characterized drug release kinetics by fitting standard release to Higuchi, zero-order and first-order models.

3.2.1. First Dissolution Tests at pH 6.8 were carried out with 0.5-1.41 mm Granules for FS, NM, FS + NM, CFS, CNM and CNM + CFS samples

The use of time-dependent polymers (Eudragit® NM) and pH-dependent polymers (Eudragit® FS) and the use of combinations of Eudragit® time- and pH-dependent polymers have been used in different works to improve colonic release [43-44]. Moreover, the absence of burst effects in the diferent Metocel®/Eudragit® granules, justified the efficient interpenetration of MLX within the network [45].

Figure 4. Release profiles at pH 6.8 for MLX raw material and multiparticulate colon systems of 0.50-1.41 mm diameter size: FS, NM, NM + FS, CFS, CNM and CNM + CFS.

As it can be seen in Figure 4, CNM granules achieved the maximum percentage released of MLX at 8 h (88.77 ± 3.46%), yielding more than the reference active ingredient, the pure MLX (78.10 ± 3.01% at the same time), followed by the CFS, with 66.34 ± 2.12% and CNM + CFS with 61.86 ± 2.60% at 8 h. Different studies have shown how the improvement in drug solubility in a pH zone (6.5-7) increases drug solubility at the colon site [25,46]. CNM formulation had the lowest “burst effect”, thus retaining better MLX for a possible enteral use. As expected, the lowest results were obtained for samples without the disintegrating material (Metolose®), FS, NM and FS+NM.

Based on the results obtained, the CNM granules were selected for further studies, followed by the CFS, and CNM + CFS, since it is at this pH (colonic) that we would be interested in starting the release of the MLX. Han and Choi [47] studied the pharmacokinetics of MLX with ethanolamine salts through their dissolution profiles at pH 1.2 and 6.8 in vitro and in vivo tests in rats, showing that the formulations with the higher dissolution at pH 6.8 also had the faster absorption of the MLX in rats, so it was expected for the CNM granulate. Therefore, the granules, CFS, CNM and CNM + CFS were prepared for a dissolution rate study at different pHs, in an in vitro simulated gastrointestinal environment.

To evaluate the influence of MLX within the multiparticulate systems, drug release kinetics were determined for the different formulations. These granules are composed of a matrix based mainly on lactose, a hydrophilic diluent, and in some systems, it is also made up with a proportion of Metolose®, disintegrant, and they are coated with, either Eudragit® NM, Eudragit® FS or a combination thereof. Drug may interact differently with the various components of the matrix. Table 2 shows the Korsmeyer-Peppas, Higuchi and first-order kinetic models of formulations: FS, NM+FS, NM, CNM + CFS and CNM. The Korsmeyer-Peppas kinetic model exhibits anomalous (non-Fickian) dissolution with n values ranging from 0.60 to 0.82, for most formulations excepting NM+FS, NM and CFS, suggesting that the rate of water uptake into the matrix of granules and MLX release were controlled by diffusion through the system structure and dissolution and swelling/erosion processes [48]. The absence of burst effects in both, CNM and CNM + CFS granules, justified the efficient interpenetration of MLX within the network [45]. According to the Korsmeyer-Peppas model, the release rate and release mechanism for CFS granules are governed by a complex quasi-Fickian release mechanism [49]. Moreover, for both systems NM+FS and NM, diffusional exponent (n) values greater than 0.89 characterize Super Case II transport [50] which is controlled by polymer chain relaxation, and erosion.

Table 2. Korsmeyer-Peppas, Higuchi, and first-order kinetic models applied for MLX release from MLX raw material and multiparticulate colonic systems CNM + CFS, CFS and CNM.

Formulations

Korsmeyer-Peppas

Higuchi

First-order

n

r2

n

r2

N

r2

FS

0,8287

0,9799

1,8851

0,9072

-0,0009

0,9023

NM + FS

1,2865

0,944

1,8552

0,9966

-0,0008

0,9943

NM

0,9818

0,9767

2,2698

0,9931

-0,0012

0,9989

CNM + CFS

0,6754

0,9337

3,3662

0,9906

-0,0023

0,9706

CFS

0,3425

0,8971

2,6235

0,8756

-0,0021

0,8412

MLX

0,6073

0,9603

4,0304

0,946

-0,0037

0,935

CNM

0,8114

0,9871

5,1589

0,9947

-0,0056

0,9995

Not all formulations fit to the zero-order kinetic model (data not shown). However, the first-order kinetic model of multiparticulate colonic systems with Eudragit® NM showed the highest r2 values, and CNM formulation was the best fitting granulate (0.995), followed by the Higuchi kinetic model (best r2 values for Eudragit® NM systems), as presented in Table 2. These results indicate that MLX release is controlled by a combination of diffusion, together with rapid swelling and erosion processes from the lactose, MC and Eudragit® NM matrix. These multiparticulate colonic systems showed kinetics of MLX dissolution that fit non-Fickian transport rather than Case II transport [48].

3.2.2. Dissolution rate of granules of size 0.35-1.50 mm at different pHs (6.8 and 7.4)

Time-dependent polymers with Metolose® such as CNM and CNM + CFS could have less influence on the microenvironment pH induced by the solubilized MLX, which is a key factor controlling drug release [32]. However, the solubilization of Eudragit® FS is pH dependent, the effect of the micro pH environment created by drug could affect their dissolution profiles [46].

Afterwards, the widest range of particle diameter was used (0.350-1.50 mm) for the selected formulations: CFS, CNM and CNM + CFS, comparing them to MLX raw material, were analyzed separately at pH 6.8 (Figure 5) and at pH 7.4 (data not shown).

At pH 6.8, it was evident that formulations with Eudragit® NM did not exhibit burst effect. CNM granulate at 8 h managed to release something more (98.48 ± 2.28%) than the pure MLX (96.14 ± 10.76%) and yielding in a more sustained manner than the rest of the formulations (Figure 5), having a similar profile to the CNM + CFS granules. However, at 180 min CNM + CFS system released slightly more MLX (59.23 ± 8.04%) than CNM formulation (56.19 ± 4.24%). The granulate that released the most MLX at initial times was CFS, achieving a 44.02 ± 0.55 % and 75.90 ± 1.44 % at 30 min and 60 min respectively, but also the CFS profile was uncontrolled and possessed a great burst effect. Following oral administration, these systems are subjected to challenges from the harsh and hostile environment of the GI tract before reaching the colon. For instance, the unwanted burst drug release in the stomach and the small intestine before reaching the colon is an intrinsic limitation of colon-targeted drug delivery systems [51]. This may be due to both factors: disintegrating effect of Metolose®, which between the intermolecular spaces of the cellulose allows the passage of water and, with it, a greater release of drug [52] and using Eudragit® FS, a pH dependent polymer, which allows drug release to be reduced at pH greater than 7.0 [46]. These data indicated that a high release of MLX from the CFS formulation at pH 6.8 decrease its chance of success in reaching the colon and exerting its action there. We are looking for a formulation that releases in the colon, so we could select the formulation that contains a time dependent Eudragit®, which deliver at the desired level of the GIT.

Table 3 shows the Korsmeyer-Peppas, Higuchi and first-order kinetic models of formulations: CFS, CNM + CFS and CNM. The Korsmeyer-Peppas kinetic model for the three formulations, with this large range of particulate size, exhibits n values of about 1.0, indicating that polymer relaxation and polymer dissolution are the dominant mechanism, which agrees with Eudragit® FS dissolution characteristics [53]. The Higuchi kinetic model of CNM multiparticulate colonic system showed the highest r2 values, followed by the first-order kinetic model, as presented in Table 3. The kinetic models’ results revealed that MLX release is controlled by a combination of diffusion (major), together with dissolution and erosion processes from Metolose® and Eudragit® (minor). The wide particle size range slightly affects the fit of the MLX release to the different kinetic models, as seen in Tables 2 and 3.

Assay carried out in pH 7.4 medium (data not shown) exhibited that the CFS granulate with pH dependent and permeability characteristics continues to be the formulation that yield faster the MLX followed by the combination CNM + CFS. On the other hand, the CNM granulate which shows a low permeability and pH independence achieved a sustained release profile that is adequate for an enteral administration.

Figure 5. Release profiles of multiparticulate colonic systems of 0.35-1.50 mm diameter size (CNM, CFS and CNM + CFS) and pure MLX in pH 6.8 medium.

Table 3. Korsmeyer-Peppas, Higuchi, and first-order kinetic models applied for MLX release from MLX raw material and multiparticulate colonic systems CNM + CFS, CFS and CNM at pH 6.8.

Formulations

Korsmeyer-Peppas

Higuchi

First-order

n

r2

n

r2

N

r2

CNM + CFS

1.1911

0.9573

58765

0.9985

-0.0082

0.868

CFS

0.9702

0.9871

4.2177

0.8693

-0.0097

0.9824

MLX

0.7534

0.9464

4.1776

0.9135

-0.0058

0.9725

CNM

0.96

0.9956

5.6754

0.9884

-0.0069

0.9843

The kinetic parameters of these multiparticulate colonic systems at this pH (7.4) could not be evaluated due to their fast dissolution.

3.2.3. Dissolution Rate of MLX at pH 1.2, 6.8 and 7.4, of 0.85-1.00 mm Diameter Granules of CFS, CNM and CNM + CFS with Buffer Changes (simulating the gastrointestinal environment)

The use of dissolution studies with pH changes at 1.2, 6.8 and 7.4 are more suitable to determine the dissolution rate of drug colon targeting systems [43-44]. Finally, the selected multiparticulate colonic systems were tested in three different media of pH 1.2, 6.8 and 7.4, for simulating their transit through the gastrointestinal tract.

At pH 1.2 (see Figure 6.A), it was observed that MLX raw material was the only sample that began to dissolve, reaching a 3.63 ± 1.98% at 2 h of testing. Therefore, the other formulations do not begin to release at the pH of the stomach and may reach other parts of the gastrointestinal tract intact. The results show that at 8 h, in this simulated model of the gastrointestinal tract, CNM granulate continues to be the one that has yielded the most, with 90.32 ± 2.34 %; but, after passing through the acid medium, the next best is the CNM + CFS, with 52.42 ± 3.25% and finally the CFS (32.24 ± 4.83%) contrary to what happened when the dissolution rate was determined only at pH 6.8. On the other hand, the size of the granules affected their dissolution, achieving a higher percentage of MLX dissolved at 24 h in the granules that presented greater specific surface [54] except in the case of the CFS granule, but the order of dissolution of the multiparticulate systems was not altered at 8 h, since it was still the CNM (90.32 ± 2.34%) that released the most MLX at that time, compared to the pure MLX with a 46.96 ± 4.91% achieved at 8 h. These results showed that CNM formulation continues to present a low burst effect, possessing a sustained release, which could allow a high release of the active ingredient at the pH of the site of action and at times consistent with intestinal transit. In addition, after successive filtrations and medium changes, it was observed that the three formulations (CFS, CNM and CNM + CFS multiparticulate colonic systems) maintained their full structure, as granules.

Figure 6. (A) Dissolution rate of MLX with buffer changes (pH 1.2, 6.8 and 7.4) for MLX raw material and 0.85-1.00 mm size diameter granules of CNM, CFS and CNM + CFS. (B) Higuchi kinetic model applied for MLX release from MLX raw material, multiparticulate colonic systems CNM + CFS, CFS and CNM. (C) First-order kinetic model applied for MLX release from MLX raw material, multiparticulate colonic systems CNM + CFS, CFS and CNM.

To evaluate the influence of MLX within the granulate matrix, drug release kinetics were determined for the different formulations. Figures 6.B and 6.C show the Higuchi and first-order kinetic models, respectively, for multiparticulate systems: CNM, CNM + CFS and CFS. Because of the pH changes, these results could not be fitted to the Korsmeyer-Peppas kinetic model.

CNM formulation presented highest r2 values in the first-order (0.9661) and in Higuchi (0.9961) kinetic models, respectively (Figures 4.b and 4.c); followed by CNM + CFS (0.9651 and 0.9729) and finally CFS obtained the worst results for both first-order (0.9585) and Higuchi (0.9476) kinetics, respectively. In view of these results, we can conclude that in formulations containing Eudragit® NM, a time-dependent polymer, and Metolose®, the predominant mechanism of drug release is diffusion [55] until the Eudragit® NM polymer becomes more porous and dissolves. From then on, the phenomenon of swelling and erosion of the Metolose® begins to occur, which controls the release of MLX to a lesser extent.

3.3. Scanning Electron Microscopy (SEM)

The Scanning Electron Microscopy (SEM) method was used to study the surface and morphological characteristics of different samples: Metolose®, Eudragit® FS and Eudragit® NM in their pure states, MLX raw material, CNM granulate (selected for its adequate dissolution rate characteristics) composed of MLX, lactose, Metolose® and Eudragit® NM, depending on the sample, and its blank formulation (which had lactose instead of the active ingredient) with four successive granulations. Moreover, we have performed SEM surface images which shall give information about changes in the surface morphology during dissolution [56], simulating the gastro-intestinal conditions: pH 1.2 (gastric pH), pH 6.8 (small intestine pH) and pH 7.4 (colon pH).

Scanning electron micrographs of MLX raw material, at different magnifications (data not shown) exhibit the morphological characteristics of the drug, whose structure is in the form of recognizable cubic crystals [57] with a smooth surface of approximately 3.5 μm. This MLX characteristic structure can be well observed in the multiparticulate system analyzed at 200X magnification with four granulations (Figure 7.A). However, MLX was not seen, since drug is not present, in the blank granules, as in the case of the blank formulation with four successive granulations (Figure 7.B).

This detail is better appreciated in the micrographs obtained with a magnification of 1500× (Figures 7.C and 7.D), verifying that MLX is not seen in the blank sample (Figure 5.D), observing the presence of lactose only, easily distinguishable for its large crystals. These changes were also observed in other type of blank formulations [58].

In the micrographs obtained at a magnification of 20× (Figures S1), it was also possible to identify the approximately particle morphology, size, and shape by SEM. Benavent C et al [59] performed an SEM analysis identifying its surface and an approximate particle size range for different solid dispersions. In Figure S1.A (CNM formulation), it is shown irregular particles with a matrix structure and a rough surface (0.51-0.24 mm) and in Figure S1.B (blank CNM formulation) similar structures are shown with a slight larger particle size (0.67-0.27 mm).

In Figure S2 it is shown the surface morphology of Metolose®, Eudragit® FS and Eudragit® NM in their pure states. As seen in Figure S2.A, methylcellulose (Melose®) appears as irregular flake fibers, also described by other authors [56]. On the other hand, Eudragit® FS and NM displayed a very smooth film surface, with some scales of different thickness, this smooth surface has been previously reported in formulations with Eudragit® [60].

Figure S3 displays scanning electron micrographs of the surfaces of multiparticulate systems CFS, CNM + CFS and CNM, after 2 h at pH 1.2; 3 h at pH 6.8 and 3 h at pH 7.4. After 2 h in simulated gastric fluid (pH 1.2) the surface structure of all three formulations remained very similar to the initial time (t0), presenting a very rough surface (see Figure 7). However, after being in an alkaline medium (pH 6.8), the three multiparticulate systems showed a more porous structure. Similar studies were carried out by several authors [53, 56, 61]. Therefore, SEM images (pH 6.8) let assume that the dissolution happens through pores that are generated by a combination of drug diffusion [56] and polymers swelling / eroding processes [53]. Meanwhile, after being 3 h in pH 7.4 CFS and CNM + CFS systems showed a smoother structure, this could be related to a rapid polymer disintegration. In contrast, the CNM formulation still had a rough surface, maintaining its structure. This result may be related to its more controlled dissolution rate. Similar results were obtained by Mehta et al. [61] in Eudragit® S-100 coated naproxen matrix tablets for colon-targeted drug delivery system.

The SEM results allowed us to consider that the granulation process managed to effectively create a consistent polymeric matrix structure in CNM formulation, thus achieving a sustained release throughout the gastrointestinal tract.

Figure 7. Scanning electron micrographs of CNM multiparticulate system and its blank formulation at a magnification of 200× and the scale bar is equal to 150 µm: (A) CNM multiparticulate system; (B) blank CNM formulation; Scanning electron micrographs of CNM multiparticulate system and its blank formulation at a magnification of 1500× and the scale bar is equal to 20 µm:(C) CNM formulation and (D) blank CNM formulation.

3.4. X-Ray Powder Diffractometry (XRPD)

The objective of X-ray powder diffractometry (XRPD) was to identify the changes in the crystalline structure of the MLX raw material and the possible dilution effect due to the addition of different excipients such as lactose, Eudragit® and Metolose® incorporated into the formulation CNM. Meanwhile the grazing incidence X-ray diffraction (GID) was to perform a better evaluation of the surface of the selected formulation (CNM) due to its adequate MLX release profile [38].

Figure 8.A shows the differences between the diffractograms obtained by XRPD and GID techniques of the CNM granulate containing MLX. The classical XRPD method shows the main characteristic peaks of MLX in the CNM granulate. Likewise, when using the GID technique, it is also possible to distinguish the majority peaks of the MLX at angle 2 (θ) positions of 8.5 ± 0.2° and 14.5 ± 0.2°. In contrast, as expected, the XRPD diffractogram of the CNM placebo pellet does not show any peak characteristic of MLX. In addition, it is shown that the pure MLX is presented, in its polymorphic form III, which is characterized by peaks at the positions of 8.5 ± 0.2°, 10.7 ± 0.2°, 12, 7 ± 0.2° and 14.54 ± 0.2° [62] and characteristic peaks at 16.4± 0.2° and 17.4 ± 0.2° for lactose [63]. However, with the GID technique in the CNM formulation, a peak appears at position 28.4 ± 0.2° with an elevated intensity, which corresponds to a particularly crystalline component of the granulate with large crystals, possibly due to lactose. In all samples analyzed by classic XRPD technique, except for the MLX raw material, an identical peak is seen in this position, regardless of whether they contain MLX. To identify that peak, it is decided to analyze all the excipients of CNM formulation (see figure 8.B), to try to identify it. Thus, lactose, Metolose® and Eudragit® NM were analyzed. The latter was previously allowed to dry in a petri dish, and then pulverizing it to be assayed by the XRPD technique. Diffractograms of Metolose® and Eudragit® NM presented halos that can be associated with a semi-crystalline state, but mostly amorphous. Both polymers Metolose® and Eudragit® could present a modest degree of crystallinity that has been previously described [64, 65]. Lactose presented a highly crystalline pattern, with a peak at the same position (28.4 ± 0.2°) as in both XRPD and GID diffractograms of CNM formulation. The XRPD results of CNM formulation demonstrated that after the granulation process, the original structure of all components is maintained.

Figure 8. (A) Comparison between powder X-ray diffraction (XRD) and grazing X-ray diffraction (GID) patterns of CNM (GID); CNM (XRPD); blank CNM (XRPD) and pure MLX (XRPD). (B) Comparison between powder X-ray diffraction (XRD) and grazing X-ray diffraction (GID) patterns of CNM (GID); Lactose (XRPD); blank CNM (XRPD); Metolose® (XRPD) and Eudragit® NM (XRPD).

  1. Conclusions

Different multiparticulate MLX systems have been designed for colonic administration based on lactose, methyl cellulose (Metolose®), Eudragit®NM, FS and combinations of both polymers. CFS and CNM multiparticulate systems with Metolose® showed a significant increase (p < 0.05) in solubility, compared to FS and NM formulations. Under conditions like gastrointestinal transit (pHs 1.2, 6.8 and 7.4), CNM formulation showed the best MLX release characteristics, with a more sustained profile compared to the other formulations studied. CNM multiparticulate system presented highest r2 values in both the first order (0.9661) and in Higuchi (0.9961) kinetic models, respectively; followed by CNM + CFS (0.9651 and 0.9729) and finally CFS obtained the worst results for both first order (0.9585) and Higuchi (0.9476) kinetics, respectively. In view of these results, we can consider that in formulations containing Eudragit®NM, a time-dependent polymer, and Metolose®, the predominant mechanism of drug release is diffusion followed by swelling/eroding phenomena. SEM results showed that the granulation process managed to effectively coat the MLX in CNM granulate, thus achieving a sustained release throughout the gastrointestinal tract. The X-ray diffraction results of CNM formulation demonstrated that after the granulation process, the original structure of all components is maintained. Finally, colonic release formulation of MLX has been achieved, which reaches the large intestine mostly intact, with a sustained release along the intestinal tract due to the use of Eudragit® NM in four successive granulations and a high release of active ingredient at the site of action, due to the presence of Metolose®. It also has an adequate shape and size, suitable for uniform dosage and possibly appropriate for the treatment of diseases involving colonic inflammation such as ulcerative colitis, Crohn's disease, and colon cancer among others.

Author Contributions: Conceptualization, C.A.-A. and P.T.-I.; methodology, C.T.-S, C.A.-A. and P.T.-I.; investigation, E.N.-R. and and Z.-D.; resources, C.A.-A and P.T.-I.; writing—original draft preparation, E.N.-R., C.A.-A., and P.T.-I.; writing—review and editing, C.A.-A., M.A.P., C.T.-S. and P.T.-I; visualization, C.A.-A., M.A.P., C.T.-S., and P.T.-I..; supervision, C.A.-A. and P.T.-I.; funding acquisition, C.A.-A. and P.T.-I.. All authors have read and agreed to the published version of the manuscript.

Funding: This research was funded by the projects Ministerio de Ciencia e Innovación [MICINU, ref. RTI2018-093940-B-100] and UCM [Research Group 910939].

Acknowledgments: we acknowledge CAI Difracción de rayos-X of UCM for X-ray diffractometry measurements, and the ICTS Centro Nacional de Microscopia Electrónica (UCM) for SEM images.

Institutional Review Board Statement: Not applicable.

Informed Consent Statement: Not applicable.

Conflicts of Interest: The authors declare no conflict of interest.

References

  • Cheng, H.; Huang, S.; Huang, G. Design and application of oral colon administration system. J. Enzyme Inhib. Med. Chem. 2019, 34, 1590-1596. doi.org/10.1080/14756366.2019.1655406.
  • Newton, A.; Prabakaran L.; Jayaveera, K. Pectin-HPMC E15LV vs pH sensitive polymer coating films for delayed drug delivery to colon: a comparison of two dissolution models to asses colonic targeting performance in-vitro. J. Appl. Res. Nat. Prod. 2012, 5, 1 – 16.
  • Philip, A.K.; Philip, B. Colon targeted drug delivery systems: a review on primary and novel approaches. Oman Med. J. 2010, 25(2), 79–87. doi: 10.5001/omj.2010.24.
  • Chourasia, M.; Jain, S. Pharmaceutical approaches to colon targeted drug delivery systems. J. Pharm. Pharm. Sci. 2003, 6(1), 33-66.
  • Basit, A.; Bloor, J. Perspectives on colonic drug delivery business briefing. Pharmatech 2003, 185–190.
  • Asghar, L.; Chandran, S. Multiparticulate formulation approach to colon specific drug delivery: current prospective. Pharm. Pharm. Sci. 2006, 9, 327 – 338.
  • Friend, R. New oral delivery systems for treatment of inflammatory bowel disease. Drug Deliver. Rev. 2004, , 57, 247 - 265.
  • Arriagada, F.; Ugarte, C.; Günther, G.; Larraín, M. A.; Guarnizo-Herrero, V.; Nonell, S.; Morales, J. Carminic acid linked to silica nanoparticles as pigment/antioxidant bifunctional excipient for pharmaceutical emulsions. Pharmaceutics 2020, 12(4), 376. doi.org/10.3390/pharmaceutics12040376.
  • Marks, S.; Schneider, J.; Keely, S. Advances in oral nano-delivery systems for colon targeted drug delivery in inflammatory bowel disease: selective targeting to diseased versus healthy tissue. Nanomedicine 2015, 11, 1117 – 1132. doi.org/10.1016/j.nano.2015.02.018.
  • Bak, A.; Ashford M.; Brayden, D. Local delivery of macromolecules to treat diseases associated with the colon. Drug Deliv. Rev. 2018, 2 - 27, 136 -137. doi.org/10.1016/j.addr.2018.10.009.
  • Naeem, M.; Choi, M.; Cao J.; Lee, Y.; Ikram M.; Yoon, S.; Lee, J. Moon; H. Kim, M.; Jung, Y.; Yoo, J. Colon-targeted delivery of budesonide using dual pH-and timedependent polymeric nanoparticles for colitis therapy. Drug Des. Devel. Ther. 2015, 9, 3789 – 3799. doi: 10.2147/DDDT.S88672.
  • Maroni, A.; Moutaharrik, S.; Zema, L.; Gazzaniga, A. Enteric coating for colonic drug delivery: state of the art. Expert Opin. Drug Deliv, 2017, 14(9), 1027 – 1029. doi.org/10.1080/17425247.2017.1360864.
  • Rashid, N.M.; Kaur, V.; Hallan, S.; Sharma, S.; Mishra, N. Microparticles as controlled drug delivery carrier for the treatment of ulcerative colitis: a brief review. Saudi Pharm. J. 2016, 24, 458-472. doi.org/10.1016/j.jsps.2014.10.001.
  • Naik, J.B.; Waghulde, M.R. Development of vildagliptin loaded Eudragit® microspheres by screening design: in vitro evaluation. Pharm. Investig. 2018, 48, 627 – 637. doi.org/10.1007/s40005-017-0355-3.
  • Hua, S. Orally administered liposomal formulations for colon targeted drug delivery. Pharmacol. 2014, 5, 138, 1-4. doi.org/10.3389/fphar.2014.00138.
  • Patel, M. Cutting-edge technologies in colon-targeted drug delivery systems. Expert Opin. Drug Deliv. 2011, 8, 1247 – 1258.
  • Raish, M.; Kalam, M.A.; Ahmad, A.; Shahid, M.; Ansari, M.A.; Ahad, A.; Ali, R.; Bin Jardan, Y.A.; Alshamsan, A.; Alkholief, M.; Alkharfy, K.M.; Abdelrahman, I.A.; Al-Jenoobi, F.I. Eudragit-coated sporopollenin exine microcapsules (SEMC) of phoenix dactylifera l. of 5-fluorouracil for colon-specific drug delivery. Pharmaceutics 2021, 13, 1921.
  • Jain, V.; Singh, R. Development and characterization of Eudragit RS100 loaded microsponges and its colonic delivery using natural polysaccharides. Acta Pol. Pharm: Drug Res. 2010, 67, 407 – 415.
  • Chourasia, M.K.; Jain, S.K. Polysaccharides for colon targeted drug delivery. Drug Deliv. 2004, 11(2), 129-48. doi: 10.1080/10717540490280778.
  • Maurer, A.H. Gastrointestinal motility, part 2: small-bowel and colon transit. Nuc.l Med. Technol. 2016, 44(1), 12-8. doi: 10.2967/jnumed.113.134551.
  • Guo, Y.; Zong, S.; Pu, Y.; Xu, B.; Zhang, T.; Wang, B. Advances in pharmaceutical strategies enhancing the efficiencies of oral colon-targeted delivery systems in inflammatory bowel disease. Molecules 2018, 23, 1622. doi.org/10.3390/molecules23071622.
  • Vemula, S.K.; Veerareddy, P.R. Development, evaluation and pharmacokinetics of time-dependent ketorolac tromethamine tablets. Opin. Drug Deliv. 2013, 10, 33 - 45. doi.org/10.1517/17425247.2013.743528.
  • Park, H.; Jung, H.; Ho M.; Lee, D.R.; Cho, H.R.; Choi, Y.S.; Jun, J.; Son, M.; Kang, M.J. Colon-targeted delivery of solubilized bisacodyl by doubly enteric-coated multiple-unit tablet. J. Pharm. Sci. 2017, 102, 172 – 179. doi.org/10.1016/j.ejps.2017.03.006.
  • Cheng, H.; Huang S.; Huang, G. Design and application of oral colon administration system. Enzyme Inhib. Med. Chem. 2019, 34, 1590 – 1596. doi.org/10.1080/14756366.2019.1655406.
  • Patel, M.; Amin, A. Formulation and development of release modulated colon targeted system of meloxicam for potential application in the prophylaxis of colorectal cancer. Drug Deliv. 2011, 18, 281 - 293.
  • Ochi, M.; Inoue, R.; Yamauchi, Y.,; Yamada, S.; Onoue, S. Development of meloxicam salts with improved dissolution and pharmacokinetic behaviors in rats with impaired gastric motility. Res. 2013, 30, 377–386. doi.org/10.1007/s11095-012-0878-2.
  • Tsujii, M. COX-2 inhibitor and colon cancer. Gan to Kagaku Ryoho 2001, 28, 1799 – 1805.
  • Tsubouchi, Y.; Mukai, S.; Kawahito, Y.; Yamada, R.; Kohno M.; Inoue, K.; Sano, H. Meloxicam inhibits the growth of non-small cell lung cancer. Anticancer Res. 2000, 20, 2867 – 2872.
  • Lee, S.H.; Bajracharya, R.; Min, J.Y.; Han, J.W.; Park, B.J.; Han, H.K. Strategic approaches for colon targeted drug delivery: an overview of recent advancements. Pharmaceutics 2020, 15, 12(1), 68. doi: 10.3390/pharmaceutics12010068.
  • Palugan, L.; Cerea, M.; Zema, L.; Gazzaniga, A.; Maroni, A. Coated pellets for oral colon delivery. Drug Del. Sci. Tech. 2015, 25, 1 – 15. doi.org/10.1016/j.jddst.2014.12.003.
  • Woraphatphadung, T.; Sajomsang, W.; Gonil, P.; Treetong, A.; Akkaramongkolporn, P.; Tanasait Ngawhirunpat, Praneet O. pH-responsive polymeric micelles based on amphiphilic chitosan derivates: Effect of hydrophobic cores on oral meloxicam delivery. J. Pharm. 2016, 497 (1-2), 150 – 160. doi.org/10.1016/j.ijpharm.2015.12.009.
  • Navarro, E.; Álvarez, C.; García, J.; Torrado, S.; Torrado, S.; Torre, P. New multi-particle systems for colon-targeted meloxicam. J. App. 2016, 8, 221. doi.org/10.21065/1920-4159.1000221.
  • Ferlay, J.; Colombet, M.; Soerjomataram, I.; Dyba, T.; Randi, G;. Bettio, M.; Gavin, A.; Visser O;. Bray F. Cancer incidence and mortality patterns in Europe: estimates for 40 countries and 25 major cancers in 2018. J. Cancer. 2018, 103, 356 – 387. doi.org/10.1016/j.ejca.2018.07.005.
  • Li, J.; Lee, I.W.; Shin, G.H.; Chen, X.; Park ,H.J. Curcumin-Eudragit® E PO solid dispersion: A simple and potent method to solve the problems of curcumin. Eur J Pharm Biopharm. 2015, 94, 322-32. doi: 10.1016/j.ejpb.2015.06.002.
  • Maruyama, S.; Ando, S.; Yonemochi, E. Application of void forming index (VFI): Detection of the effect of physical properties of dry powder inhaler formulations on powder cohesion. Int J Pharm. 2020, 15, 588, 119766. doi: 10.1016/j.ijpharm.2020.119766.
  • Li, J.; Lee, I.W.; Shin, G.H.; Chen, X.; Park ,H.J. Curcumin-Eudragit® E PO solid dispersion: A simple and potent method to solve the problems of curcumin. Eur J Pharm Biopharm. 2015, 94, 322-32. doi: 10.1016/j.ejpb.2015.06.002.
  • Mašková, E.; Naiserová, M.; Kubová, K.; Mašek, J.; Pavloková, S.; Urbanová, M.; Brus, J.; Vysloužil, J.; Vetchý, D. Highly soluble drugs directly granulated by water dispersions of insoluble Eudragit® polymers as a part of hypromellose k100m matrix systems. Res. Int. 2019, 5, 8043415. doi: 10.1155/2019/8043415.
  • Vaz, G.R.; Carrasco, M.C.F.; Batista, M.M.; Barros, P.A.B.; Oliveira, M.d.C.; Muccillo, A.; Yurgel, V.; Buttini, F.; Soare, F.; Cordeiro, L.M.; Fachel, F.; Teixeira, H.; Bidone, J.; Oliveira P.; Sonvico, F.; Dora, C.L. Curcumin and quercetin-loaded lipid nanocarriers: development of omega-3 mucoadhesive nanoemulsions for intranasal administration. Nanomaterials 2022, 12, 1073. https://doi.org/10.3390/ nano12071073.
  • Gómez-Burgaz, M.; Torrado G.; Torrado, S. Characterization and superficial transformations on mini-matrices made of interpolymer complexes of chitosan and carboxymethylcellulose during in vitro clarithromycin release. J. Pharm. Biopharm. 2009, 1, 73, 130. doi: 10.1016/j.ejpb.2009.04.004
  • Lu, J.; Obara, S.; Liu, F.; Fu, W.; Zhang, W.; Kikuchi, S. Melt extrusion for a high melting point compound with improved solubility and sustained release. AAPS PharmSciTech, 2018, 19(1), 358-370. doi.org/10.1208/s12249-017-0846-6.
  • Kadota, K.; Terada, H.; Fujimoto, A.; Nogami, S.; Uchiyama, H.; Tozuka, Y. Formulation and evaluation of bitter taste-masked orally disintegrating tablets of high memantine hydrochloride loaded granules coated with polymer via layering technique. J. Pharm. 2021, 15, 604, 120725. doi: 10.1016/j.ijpharm.2021.120725.
  • Al-Hashimi, N.; Begg, N.; Alany, R.G.; Hassanin, H.; Elshaer, A. Oral modified release multiple-unit particulate systems: compressed pellets, microparticles and nanoparticles. Pharmaceutics 2018, 10(4), 176. doi.org/10.3390/pharmaceutics10040176
  • Weyna, D.; Cheney, M.; Shan, N.; Hanna M.; Zaworotko, M.J.; Sava, D.; Song, S.; Sanchez-Ramos, J.R. Improving solubility and pharmacokinetics of meloxicam via multiple-component crystal formation. Pharmaceutics 2012, 7(9) 2094–2102. doi.org/10.1021/mp300169c.
  • Naeem, M.; Bae, J.; Oshi, M.A.; Kim, M.S.; Moon, H.R.; Lee, B.L.; Jung, Yoo, Y.; J.W. Colon-targeted delivery of cyclosporine A using dual-functional Eudragit® FS30D/PLGA nanoparticles ameliorates murine experimental colitis. J. Nanomedicine 2018, 13, 1225-1240. doi.org/10.2147/IJN.S157566.
  • Turanli, Y.; Acartürk, F. Fabrication and characterization of budesonide loaded colon-specific nanofiber drug delivery systems using anionic and cationic polymethacrylate polymers. Drug Deliv. Sci.Technol. 2021, 63, 102511. doi.org/10.1016/j.jddst.2021.102511.
  • Consumi, M.; Leone, G.; Pepi, S.; Tamasi, G.; Lamponi, S.; Donati, A.; Magnani, A. Xanthan gum–chitosan: delayed, prolonged, and burst-release tablets using same components in different ratio. Polymer Tech. 2018, 37, 2936–2945. doi.org/10.1002/ADV.21965.
  • Zhang, F. Melt-Extruded Eudragit® FS-Based Granules for Colonic Drug Delivery. AAPS PharmSciTech 2016, 17, 56–67. doi.org/10.1208/s12249-015-0357-2.
  • Han, H.; Choi, H. Improved absorption of meloxicam via salt formation with ethanolamines. J. Pharm. Biopharm. 2007, 1(65), 99-103.
  • Muhammad, H.; Ghulam, A.; Shahid, S.; Muhammad, Z.; Akhtar, R.; Abdul, M.; Mehmood, K.S.; Masood. A.M. Raft-forming system for pantoprazole and domperidone delivery: in vitro and in vivo study. Bioinspired Biomim. Nanobiomaterials 2020, 9(3), 137-146. doi.org/10.1680/jbibn.19.00031.
  • Solomon, S.; Iqbal, J.; Albadarin, A.B. Insights into the ameliorating ability of mesoporous silica in modulating drug release in ternary amorphous solid dispersion prepared by hot melt extrusion. J. Pharm. Biopharm. 2021, 165, 244-258. doi.org/10.1016/j.ejpb.2021.04.017.
  • Iurckevicz, G.; Dahmer, D.; Santos, V.; Vetvicka, V.; Barbosa-Dekker, A.; Dekker, R.; Maneck, C.R.; da Cunha, M.A. Encapsulated microparticles of (1→6)-β-d-glucan containing extract of baccharis dracunculifolia: production and characterization. Molecules 2019, 3, 24(11), 2099. doi.org/10.3390/molecules24112099.
  • Naeem, M.; Awan, U.A;, Subhan, F.; Cao, J.; Hlaing, S.; Lee, J.; Im, E.; Jung, Y.; Yoo, J. Advances in colon-targeted nano-drug delivery systems: challenges and solutions. Pharm. Res. 2020, 43, 153–169. doi.org/10.1007/s12272-020-01219-0.
  • Papp, J.; Marton, S.; Süvegh, K.: Zelkó, R. The influence of Metolose structure on the free volume and the consequent metoprolol tartrate release of patches. J Biol. Macromol. 2009, 1(44), 6 – 8. doi.org/10.1016/j.ijbiomac.2008.09.014.
  • Guarnizo, V.; Torrado, C.; Torres, N.S.; Torrado, G.; Morales, J.; Torrado-Santiago, S. Study of different chitosan/sodium carboxymethyl cellulose proportions in the development of polyelectrolyte complexes for the sustained release of clarithromycin from matrix tablets. Polymers 2021, 21, 13(16), 2813. doi.org/10.3390/polym13162813.
  • Chu, K., Lee, E.; Jeong, S.; Park, E. Effect of particle size on the dissolution behaviors of poorly water-soluble drugs. Arch Pharm Res. 2012, 7(35), 1187-1195.org/10.1007/s12272-012-0709-3.
  • Mircioiu, C.; Voicu, V.; Anuta, V.; Tudose, A.; Celia, C.; Paolino, D.; Fresta, M.; Sandulovici, R.; Mircioiu, I. Mathematical modeling of release kinetics from supramolecular drug delivery systems. Pharmaceutics 2019, 21, 11(3), 140. doi.org/10.3390/pharmaceutics11030140.
  • Güres, S.; Kleinebudde, P. Dissolution from solid lipid extrudates containing release modifiers. Int J Pharm. 2011, 30, 412(1-2), 77-84. doi: 10.1016/j.ijpharm.2011.04.010.
  • Bartos, C.; Szabó-Révész, P.; Bartos, C.; Katona, G.; Jójárt-Laczkovich, O.; Ambrus, R. The effect of an optimized wet milling technology on the crystallinity, morphology and dissolution properties of micro- and nanonized meloxicam. Molecules 2016, 4, 21, 507. doi.org/10.3390/molecules21040507.
  • Tascon, E.; Torre, P.; Garcia, J.; Pena, M.A.; Alvarez, C. Enhancement of the dissolution rate of indomethacin by solid dispersions in low-substituted hydroxypropyl cellulose. Indian J. Pharm. Sci. 2019, 81(5), 824-833. doi.org/10.36468/pharmaceutical-sciences.576.
  • Benavent, C.; Torrado-Salmerón, C.; Torrado-Santiago, S.; Development of a solid dispersion of nystatin with maltodextrin as a carrier agent: improvements in antifungal efficacy against Candida spp. Biofilm Infections. Pharmaceuticals (Basel). 2021, 22, 14(5), 397. doi: 10.3390/ph14050397.
  • Mustafa, W.W.; Fletcher, J.; Khoder, M.; Alany, RG. Solid dispersions of gefitinib prepared by spray drying with improved mucoadhesive and drug dissolution properties. AAPS PharmSciTech 2022, 23, 48. https://doi.org/10.1208/s12249-021-02187-4.
  • Mehta, R.; Chawla, A.; Sharma, P.; Pawar, P. Formulation and in vitro evaluation of Eudragit S-100 coated naproxen matrix tablets for colon-targeted drug delivery system. Adv. Pharm. Technol. Res. 2013, 4(1), 31-41. doi: 10.4103/2231-4040.107498. PID: 23662280; PMCID: PMC3645356.
  • Jacon, J.; Santos, O.; Bonfilio, R.; Doriguetto, A.; de Araújo, M. Analysis of polymorphic contamination in meloxicam raw materials and its effects on the physicochemical quality of drug product. J. Pharm. Sci. 2017, 109, 347-358. doi.org/10.1016/j.ejps.2017.08.029.
  • Abu Fara, D.; Rashid, I.; Alkhamis, K.; Al-Omari, M.; Chowdhry, B.Z.; Badwan ,A. Modification of α-lactose monohydrate as a direct compression excipient using roller compaction. Drug Dev Ind Pharm. 2018, 44(12), 2038-2047. doi: 10.1080/03639045.2018.1508224.
  • Medarević, D.; Kachrimanis, K.; Djurić Z.; Ibrić, S. Influence of hydrophilic polymers on the complexation of carbamazepine with hydroxypropyl-β-cyclodextrin. J. Pharm. Sci. 2015, 78, 273-285. doi.org/10.1016/j.ejps.2015.08.001.
  • Salatin, S.; Barar, J.; Barzegar-Jalali, M.; Adibkia, K.; Alami-Milani, M.; Jelvehgari, M. Formulation and evaluation of Eudragit RL-100 nanoparticles loaded in-situ forming gel for intranasal delivery of rivastigmine. Pharm. Bull. 2020, 1, 10, 20-29. doi.org/10.15171/apb.2020.003.

SUPPLEMENTARY MATTERIAL REVISED MANUSCRIPT

Figure S1. Scanning electron micrographs of CNM multiparticulate system and its blank formulation, at a magnification of 20× and the scale bar is equal to 1.5 mm: (A) CNM multiparticulate system; (B) blank CNM formulation.

Figure S2. Scanning electron micrographs of polymers at a magnification of 200× and the scale bar is equal to 200 µm: (A) Metolose®; (B) Eudragit®FS and (C) Eudragit®NM.

Figure S3. Scanning electron micrographs of CFF, CNM + CFS and CNM multiparticulate systems, after 2 h at pH 1.2; 3h at pH 6.8 and 3 h at pH 7.4. Original magnification is 1500x and the scale bar is equal to 30µm.

Reviewer 2 Report

The authors designed a multiparticulate system of meloxicam for colonic administration in cancer or autoimmune diseases. The characterization and the release profile of the material were properly executed. Thus, The manuscript can be considered after minor adjustments.

1) The introduction section of the manuscript should be modified (especially the order of ideas). Paragraphs 1 and 2 shouldn't come earlier than paragraph 3. 

2) In between lines 97 and 99, you cited the same reference twice. I think the former is not necessary as all the idea is taken from the same reference. 

2). The conclusion part of the manuscript is too lengthy. Please organize the conclusion in a short and precise way in a single paragraph.  

Author Response

Response to Reviewer 2 Comments

First of all, we would like to thank this reviewer for his/her comments and his/her detailed review of the manuscript.

We appreciate your constructive comments below and have made revisions accordingly.

So, we have taken into account, point by point, all reviewer suggestions to rewrite the manuscript.

Point 1: The introduction section of the manuscript should be modified (especially the order of ideas). Paragraphs 1 and 2 shouldn't come earlier than paragraph 3.

Response 1: We have followed this suggestion of the reviewer, in order to clarify the text, we have reframed the Introduction section (see pages 1-4, lines 34-136).

Point 2: In between lines 97 and 99, you cited the same reference twice. I think the former is not necessary as all the idea is taken from the same reference.

Response 2: We would like to thank this referee for this comment and of course we have modified it in the corrected manuscript (see page 4 line 128).

Point 3: The conclusion part of the manuscript is too lengthy. Please organize the conclusion in a short and precise way in a single paragraph.

Response 3: According with the referee comment we have rephrased the Conclusions section in one paragraph (see page 17 lines 574-596).

Author Response

Response to Reviewer 3 Comments

First of all, we would like to thank this reviewer for his/her comments and his/her detailed review of the manuscript.

We agree with the suggestions of this reviewer that help to clarify the manuscript.

So, we have taken into account, point by point, all reviewer suggestions to rewrite the manuscript.

Point 1: The originality and novelty of using multiparticulate systems for administration in the colon is not clear from the introduction. The advantages of these systems should be discussed with respect to others investigated with meloxicam.

Response 1: We would like to remark that the novelty of the article is to develop new meloxicam (MLX) formulations to improve colonic targeting. Currently, fewer studies have considered the development of colonic formulations using multiparticulate systems like pellets or granules of MLX. In fact, MLX tablets containing Eudragit® FS 30D, as outer coating layer, were prepared for colon targeting [1]. The decline in pH from the end of the small intestine to the colon can also result in problems, lengthy lag times at the ileo-cecal junction or rapid transit through the ascending colon which can also result in poor site-specificity for enteric-coated single-unit formulations [2]. In addition, multiparticle systems hold great potential for enhancing drug targeting as well as drug uptake. Therefore, as shown by several authors, various formulations with particle size reduction may be beneficial for colon-targeted drug delivery [3], such as, MLX pellets with Eudragit® FS [4] and MLX polymeric micelles based in amphiphilic chitosan deriavatives [5]. Pellets production process is very difficult to control and also very costly, meanwhile polymeric micelles present temporal stability and additional concerns might be linked to their drug loading capacity. However, development of MLX granulates, could be a good candidate for colon targeting, because of their simple manufacturing process, good stability, homogeneous drug load and could provide a site-specific drug delivery [6].

In order to highlight the importance of using multiparticulate colonic systems of MLX we have included these advantages in the text of the Introduction section (see page 3 lines 108-123).

REFERENCES

[1] Patel, M.; Amin, A. Formulation and development of release modulated colon targeted system of meloxicam for potential application in the prophylaxis of colorectal cancer. Drug Deliv. 2011, 18, 281 - 293.

[2] Philip AK, Philip B. Colon targeted drug delivery systems: a review on primary and novel approaches. Oman Med J. 2010 Apr;25(2):79-87. doi: 10.5001/omj.2010.24. PMID: 22125706; PMCID: PMC3215502.

[3] Lee SH, Bajracharya R, Min JY, Han JW, Park BJ, Han HK. Strategic Approaches for Colon Targeted Drug Delivery: An Overview of Recent Advancements. Pharmaceutics. 2020 Jan 15;12(1):68. doi: 10.3390/pharmaceutics12010068. PMID: 31952340; PMCID: PMC7022598.

[4] Palugan, L.; Cerea, M.; Zema, L.; Gazzaniga, A.; Maroni, A. Coated pellets for oral colon delivery. J. Drug Del. Sci. Tech. 2015, 25, 1 – 15. doi.org/10.1016/j.jddst.2014.12.003.

[5] Woraphatphadung, T.; Sajomsang, W.; Gonil, P.; Treetong, A.; Akkaramongkolporn, P.; Tanasait Ngawhirunpat, Praneet Opanasopit. pH-responsive polymeric micelles based on amphiphilic chitosan derivates: Effect of hydrophobic cores on oral meloxicam delivery. Int. J. Pharm. 2016, 497 (1-2), 150 – 160. doi.org/10.1016/j.ijpharm.2015.12.009.

[6] Navarro, E.; Álvarez, C.; García, J.; Torrado, S.; Torrado, S.; Torre, P. New multi-particle systems for colon-targeted meloxicam. J. App. Pharm. 2016, 8, 221. doi.org/10.21065/1920-4159.1000221.

Point 2: Please include in line 138-139, the wavelength and the concentration range of the calibration curve used for UV validation method.

Response 2: We would like to thank this referee for this comment and of course we have included this information in the corrected manuscript in the Materials and Methods section (see page 5 lines 176-180 and pages 5 and 6 lines 187-196).

Point 3: Then, some minor corrections:

3.1- Page 2, line 45: Please replace “active principle” by “active ingredient”.

3.2- Page 2, line 50: Please write “and” before “to ensure a specific release zone...”

3.3- Page 2, line 82: Please check it, one word is missing at the beginning of line 82. For example, you could say “therefore the development of time-dependent...”

3.4-Please, increase the size of letters and numbers in -X and -X axis of Figure 1 and 4.

3.5- Please, remove the decimals in –Y axis in all figures. For example, in Figure 4A you should have 100% as you have in Figure 4B instead of 100.00%..

Response 3: According with the referee comments we performed the following corrections:

3.1- We would like to thank this referee for this comment and of course we have corrected this typing error.

3.2- According with the referee comment we have rephrased it (See page 1 lines 44).

3.3- According with the referee comment we have included the missing word (See page 2 line 89).

3.4- We have followed these suggestions of the reviewer and we have corrected Figures 1 (see Figure 3 in the corrected manuscript) and 4 (see Figure 6 in the revised manuscript).

3.5- We have followed these suggestions of the reviewer and we have corrected Y axis in all figures (see Figures 3-6 in the corrected manuscript).
